

# Applications of satellite winds for the offshore wind farm site Anholt

Tobias Ahsbahs[1], Merete Badger[1], Patrick Volker[1], Kurt S. Hansen[1], Charlotte B. Hasager[1]

[1]Department of Wind Energy, Technical University of Denmark, Roskilde, 4000, Denmark

*Correspondence to*: Tobias Ahsbahs (ttah@dtu.com)

**Abstract.** Rapid growth in the offshore wind energy sector means more offshore wind farms are placed closer to each other and in the lee of large land masses. Synthetic Aperture Radar (SAR) offers maps of the wind speed offshore with high resolution over large areas. These can be used to detect horizontal wind speed gradients close to shore and wind farm wake effects. SAR observations have become much more available with the free and open access to data from European satellite missions through Copernicus. Examples of applications and tools for using large archives of SAR wind maps to aid offshore
site assessment are few. The Anholt wind farm operated by the utility company Ørsted is located in coastal waters and experiences strong spatial variations in the mean wind speed. Wind speeds derived from the Supervisory Control And Data Acquisition (SCADA) system are available at the turbine locations for comparison with winds retrieved from SAR. The correlation is good, both for free stream and waked conditions. Spatial wind speed variations within the wind farm derived from SAR wind maps prior to the wind farm construction are found to agree well with information gathered by the SCADA
system and numerical weather prediction models. Wind farm wakes are detected by comparisons between images before and after the wind farm construction. SAR wind maps clearly show wakes for long constant fetches but the wake effect is less pronounced for short varying fetches. Our results suggest that SAR wind maps can support offshore wind energy site assessment by introducing observations in the early phases of wind farm projects.

## 1    Introduction

Europe has a total installed offshore wind capacity of 12,631 MW from 3,589 grid-connected wind turbines in 10 countries. By 2020, offshore wind is projected to grow to a total installed capacity of 24.6 GW (Wind Europe 2017). In Northern Europe much of this development is happening in the North Sea and the Baltic Sea. With an increasing amount of wind farms already erected, suitable locations with prevailing wind directions undisturbed by land or other wind farms are becoming scarce. Therefore, new wind farms are built in less favourable locations e.g. in the lee of land masses or large wind
farms. Additionally, many shore lines are not straight but have a complex geometry that is determined by peninsulas, bays and islands. The lee effect of land i.e. the horizontal wind speed gradient due to a varying distance to shore (fetch) and wind farm wakes from other wind farms both influences the wind resource. Correct prediction of the wind resource influenced by either land or adjacent wind farms or a combination of the two is a challenging problem. This study is motivated by this challenge and focuses on the Anholt offshore wind farm in the Kattegat Strait in Denmark. It involves analysis of satellite-





based Synthetic Aperture Radar (SAR) wind maps, wind turbine data, and simulation results from the Weather Research and Forecasting (WRF) model.

Winds over the ocean can be remotely sensed by satellites carrying SAR systems (Dagestad et al. 2013). SAR systems
transmit and receive microwaves and the radar backscatter signal is very sensitive to small-scale ocean waves. This scattering mechanism is diffuse and known as Bragg scattering (Valenzuela 1978). The wind is causing cm-scale waves to form on the ocean surface that are in local equilibrium with the wind speed. The wind speed at 10m height can be retrieved from SAR observations via an empirical Geophysical Model Function (GMF) (Stoffelen & Anderson 1997; Quilfen et al. 1998; Hersbach 2010) The absolute accuracy of SAR wind retrievals is low compared to high-quality measurements from
meteorological masts. The major advantages of SAR imagery, in terms of applications for wind energy, lie in the high spatial resolution and the coverage of large areas with swath widths of several hundred kilometres.

Coastal wind speed gradients have previously been quantified from SAR wind maps and compared to model simulations by Barthelmie et al. (2007) based on the very limited number of satellite samples available at the time. Ahsbahs et al. (2017)
showed that sea surface wind speeds retrieved from SAR compare well with scanning lidar wind observations as close as 1 km from the coastline. Mapping of the mean wind speed from SAR consistently shows a wind speed gradient with increasing distance from the coastline for the seas around northern Europe (Hasager et al. 2011; Hasager, Mouche, et al. 2015). At the Anholt wind farm, Peña et al. (2017) have shown strong variability of the wind speed within the turbine rows for wind directions where land is upstream. A correct prediction of this coastal gradient is desirable for optimal placement and layout
of wind farms.

Many studies of wake effects around large offshore wind farms are focused on wake interaction within the wind farms or between closely adjacent wind farms (Barthelmie et al. 2010; Hansen et al. 2012; Nygaard 2014; Hansen et al. 2015; Volker et al. 2015; Nygaard & Hansen 2016). Investigations of wind farm wake effects based on SAR wind maps revealed the
existence of extensive wakes under certain atmospheric conditions (Christiansen & Hasager 2005; Christiansen et al. 2006; Li & Lehner 2013; Hasager, Vincent, et al. 2015). The SAR wind maps contribute with information about the far-wake field, which is typically not available from other sources.

A systematic use of SAR wind maps by the offshore wind energy industry has been lacking due to three limitations: i) SAR
observations are made at the sea surface, while wind turbine rotors operate between 30 m and 250 m height; ii) SAR images have a low temporal sampling rate on the order of a few hundred images per year, depending on the location on Earth; and iii) SAR wind retrieval has required expert skills and substantial processing capabilities. These issues have been partially overcome: A method for extrapolation of mean wind speeds retrieved from SAR at 10 m above sea level to the wind turbine hub height has been developed (Badger et al. 2016) and a number of new SAR sensors have been launched in recent years,



which increases the sampling rate and ensures continuity. The access to SAR observations and derived products, such as wind maps, is eased significantly through the Copernicus programme (ref.) and its downstream services.

Numerical Weather Prediction (NWP) models offer simulations of wind speed and direction as well as other atmospheric parameters for long time series with frequent data (e.g. hourly) at several heights in the atmosphere. The WRF model (Skamarock et al. 2008) has been used to assess offshore wind resources. Good results are obtained in the open sea but in coastal regions near upstream land mass the uncertainty increases (Hahmann et al. 2015). Wind farm wakes are not resolved by NWP models unless they are explicitly parametrized (Volker et al. 2015). Engineering wind farm models like the Park model (Jensen 1983), Fuga (Ott et al. 2011), and the G. C. Larsen model (Larsen 2009) have been applied to WRF outputs (Peña et al. 2017).

Supervisory Control And Data Acquisition (SCADA) data is available from the wind turbines at Anholt and 10-minute mean wind speeds can be inferred from those measurements (hereafter SCADA wind speed). This data set gives a unique opportunity to characterize the spatial variability of the wind speed within the wind farm and it is a baseline for comparisons with wind speeds from SAR and WRF in our analyses.

The objective of this study is to demonstrate the prediction capability of SAR imagery for an offshore wind farm site where coastal wind speed gradients and wind farm wakes interact in a complex fashion. To establish confidence in the SAR wind retrievals, we first compare wind speeds from SAR and SCADA in free stream and in wake conditions. To determine whether archived SAR wind fields can predict the spatial wind speed variability at Anholt, we analyse the mean wind speed along the most Western turbine row before and after the wind farm construction. The wind farm wake effect is quantified through comparison of mean wind speeds from SAR upstream and downstream of the wind farm. Finally, the interplay between coastal wind speed gradients and wind farm wake effects is investigated through analysis of SAR wind speeds along transects perpendicular to the coastline.

The paper is structured as follows: Section 2 introduces the location, the data sets, and preprocessing used. Section 3 addresses the methods and results. In Sect. 4, we discuss implications of the presented results for wind energy projects and in Sect. 5, we conclude on the use of SAR for coastal effects and wind farm wakes.

## 2    Location & Data

This section describes the wind farm site Anholt and the data sets and pre-processing steps used for our analyses.



## 2.1 Anholt wind farm

The Anholt Offshore Wind Farm is located in the Kattegat Strait of Denmark in the waters between Djursland and the island of Anholt in an area with fairly consistent water depths of about 15 to 19 metres, see Figure 1a. The Anholt Offshore Wind Farm is approx. 20 km long and up to 8 km wide. The shortest distance to Djursland is 16 km, while there are 21 km to the island of Anholt. The Anholt wind farm consists of 111 Siemens SWT-120- 3.6 MW wind turbines with a rotor diameter of 120m with a total capacity of 400 MW constructed during 2012-2013. The internal wind turbine spacing is 5-7 rotor diameters.

## 2.2 SAR wind fields

Wind fields retrieved from two different satellite SAR missions are used in this study. Envisat ASAR from the European Space Agency (ESA) acquired images between August 2002 and April 2012 i.e. before the construction of the Anholt wind farm. The mission was followed up by a constellation of two ESA satellites, Sentinel-1A and B, from which data is available since December 2014 and April 2016, respectively. Data until May 2017 is included for this study. The entire Sentinel-1 data series is recorded after construction of the wind farm at Anholt. The Copernicus programme publishes Envisat and Sentinel-1 A/B images under an open access license, allowing for unlimited use, both for research and commercial applications.

Wind speeds are retrieved from the SAR scenes using the SAR Ocean Products System (SAROPS) (Monaldo et al. 2015). The GMF called CMOD5.N (Hersbach 2010) is chosen for the wind speed retrieval and wind directions are needed as an ancillary input for processing. We obtain the wind directions from the Climate Forecast System Reanalysis data set (CFSR[1]) during 2002-10 and from the Global Forecasting System (GFS[2]) during 2011-17. To reduce effects of random noise in the SAR imagery and to smooth out effects of longer period waves that modify the local radar incidence angle, we resample the SAR scenes to 500-m pixel size in connection with the wind retrieval processing. Hard targets like wind turbines or offshore substations cause a strong signal in SAR images. The increased backscatter signal will cause an overestimation of the retrieved wind speed and therefore, extremely bright resolution cells are filtered out of the SAR wind maps. An archive of processed wind maps from Envisat and Sentinel-1 A and B over Europe is available from DTU Wind Energy[3]. Our analyses are based on these readily available SAR wind maps.

## 2.3 SCADA data

The wind turbine power curve links the free wind speed to a power production. This wind speed (hereafter SCADA wind speed) can be derived from power and pitch combined with the power curve provided by the turbine manufacturer. The power is monotonically increasing with the wind speed between cut-in and rated power. Therefore, the wind speed can easily

---

[1] http://nomads.ncdc.noaa.gov/data.php?name=access#cfs-reanal-data
[2] http://nomads.ncdc.noaa.gov/data/gfsanl
[3] https://satwinds.windenergy.dtu.dk





be inferred for this region. From rated power to cut-out, the power is constant but the blades are pitched increasingly. For this region, the wind speed can be inferred from the pitch signal. The resulting wind speed is equivalent to the reference wind speed used to create the power curve and is treated as a measurement at hub height.

A qualification procedure has been used to eliminate periods where the wind turbines are not grid connected and are not producing power during a complete 10-minute period or have been curtailed. The remaining periods are applicable for analysis after a final examination of the power curve. Due to a lack of undisturbed mast measurements, the inflow conditions need to be derived from the operational wind turbine data themselves. The inflow reference wind direction is determined from calibrated, undisturbed selected wind turbine yaw positions on at edge of the wind farm (cf. Peña et al. (2017) for further details).

## 2.4    Numerical wind simulations

The numerical simulations used in this study were performed with WRF. The total simulated period covers 28 years from 1990 to 2017. Simulations were performed in 10-day chunks. Each individual simulation extended in total over 11 days, with the first day being disregarded as a spin-up period. The computational domain consists of three nests with an 18 km, 6 km, and 2 km grid spacing, respectively. Here the outermost domain is forced by (ECMWF) ERA-Interim Reanalysis (Dee et al. 2011) and the results of the inner-most domain have been used for the analysis. In the horizontal direction, the innermost domain extends over 854 km and 604 km in the x and y direction. In the vertical direction, 41 vertical levels with model top at 50hPa were used, with 9 levels being within 1000 m from the surface. Wind speeds at turbine hub height have been derived by logarithmic interpolation between two model levels.

The most relevant physics parametrizations in the model set-up, are the Yonsei University Scheme (YSU) Planetary Boundary Layer (PBL) scheme (Hong et al. 2006) and the MM5 similarity surface-layer scheme. Sea surface temperatures from NOAA/NCEP are used (Reynolds et al. 2010). Further details of the model set-up and its validation are given in (Peña & Hahmann 2017). WRF wind directions representative for the wind farm domain are calculated by averaging wind direction at the same locations as for the SCADA derived wind direction.

## 3    Methods & Results

Four different methods are applied to analyse SAR wind fields around the Anholt wind farm. These are listed in Table 1 together with time periods for SCADA, SAR, and WRF data used in the respective analysis. The SCADA winds are used as reference measurements. Due to the complex shape of the coastline, averaged wind speeds can show strong gradients in two directions. To distinguish them, we choose to call wind speeds changing with the distance to shore wind speed gradients and changes along the turbine rows wind speed variability. For SAR based wake studies in Sect 3.3 and 3.4 we assume that turbines are operational.





**Table 1: Overview of the data sets and time periods used for the analysis. "Wind Direction" shows what data has been used to select the SAR wind fields.**

| Analysis | SCADA wind | SAR wind | WRF wind speed | Wind Direction |
|---|---|---|---|---|
| 3.1 Comparison of wind speeds from SAR and SCADA | 12.2014 – 06.2015 | 12.2014 – 06.2015 | – | SCADA |
| 3.2 Wind speed variability along Row A | 01.2013 – 06.2015 | 08.2002 – 04.2012 | 01.2002 – 12.2012 | SCADA/WRF |
| 3.3 Wind farm wakes from SAR | – | 08.2002 – 04.2012 12.2014 – 05.2017 | – | WRF |
| 3.4 Wind farm wakes and gradients | 01.2013 – 06.2015 | 08.2002 – 04.2012 12.2014 – 05.2017 | – | WRF |

### 3.1 Comparison of wind speeds from SAR and SCADA

5 Comparisons between SAR wind speeds and SCADA winds are carried out upstream (free stream conditions) and downstream (wake conditions) of the wind turbines at Anholt. SAR wind maps at a resolution of 500m need to be further averaged in order to better represent the wind conditions, which are measured as 10-minute means at the turbine locations (Christiansen & Hasager 2005). SAR wind speeds at the turbine locations are contaminated by reflection of the wind turbines. Instead we choose to average all resolution cells that fall in a hexagonal shape for the averaging that extends from

10 600m to 2600m from the turbine with a maximum width of 1200m. It is aligned with the wind direction in order to consider SAR resolution cells directly upstream or downstream of the turbines – indicated by the grey areas labelled "upstream" and "downstream" in Figure 1b. Wind directions from SCADA are used for the directional alignment. Comparisons are done for rows A, P and 1 on the edge of the wind farm for wind direction ranges shown in Table 2.



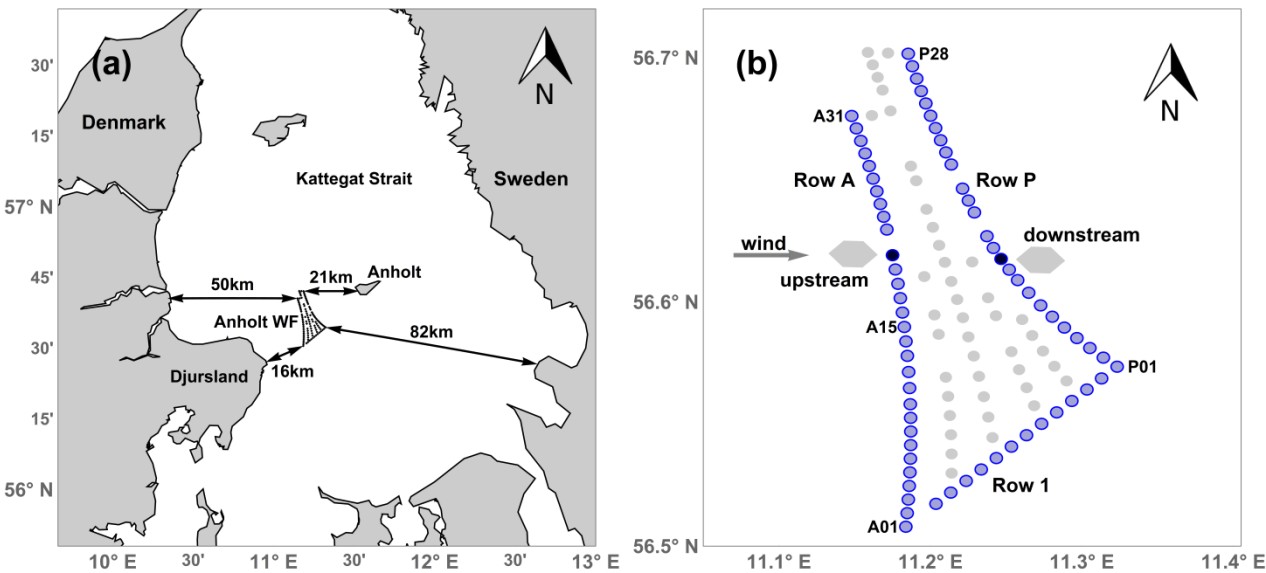

**Figure 1: a) Position of the Anholt wind farm (Anholt WF) and distances to the coast. b) Sketch of the Anholt wind farm where turbines in rows A, P and 1 are used for comparisons are marked in blue. The remaining turbines are located at the grey circles. The grey hexagons are examples of areas used for extracting the average SAR wind speeds upstream and downstream of the wind turbines for an example wind direction of 270°. The turbines used for comparisons in this example are marked in black.**

Table 2: Wind direction ranges for SAR/SCADA comparisons for upstream and downstream comparisons.

|  | Row A | Row 1 | Row P |
|---|---|---|---|
| upstream | 210° to 330° | 80° to 210° | 10° to 100° |
| downstream | 30° to 150° | 260° to 30° | 190° to 290° |

In the absence of reliable stability measurements, logarithmic wind profiles are used to extrapolate SAR wind speeds up to hub height at 81.6m. Extrapolation of SCADA winds from hub height down to 10 m where the SAR winds are retrieved is included as well, since references on SAR wind speed accuracy are given for this height. A wind speed dependent roughness length is applied in connection with the extrapolations using Charnock's relation and the Charnock parameter (Grachev & Fairall 1996). In absence of a better option, logarithmic profiles are assumed for comparisons downstream. The following results are based on SAR wind maps from 47 Sentinel-1A images collocated with the available SCADA data.

### 3.1.1 Upstream

Comparisons at hub height upstream of the wind turbines are shown in Figure 2a. SCADA wind speeds at hub height range from 4 m/s to 20 m/s covering most of the range of wind turbine operation. Comparisons with SAR wind speeds yield a mean bias of -0.16 m/s with a slight tendency of SAR to estimate higher winds. The correlations coefficient ($R^2$) of the




linear fit through the origin is 0.96, the slope of the fit is close to one, and the RMSE is 2.33 m/s. Wind speeds at 10m in Figure 2b are generally lower and the RMSE of the comparison is lower due to this (1.80 m/s).

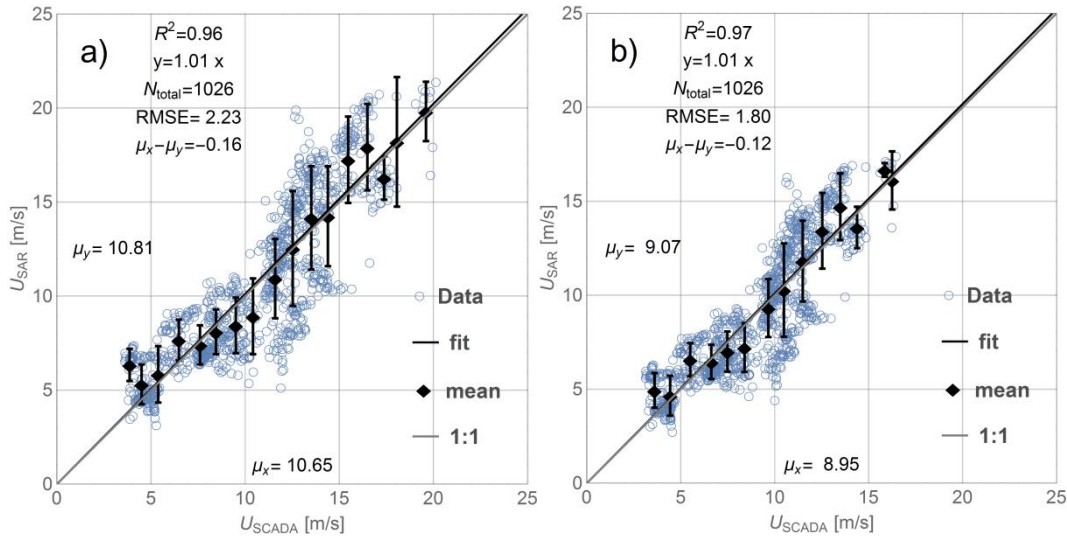

**Figure 2: Comparison between SCADA derived wind speeds ($U_{SCADA}$) and SAR derived wind speed ($U_{SAR}$) upstream of the wind**
**turbines: a) for the turbine hub height (81.6 m), b) for the reference height 10 m.**

The low bias, good correlation and slopes close to one suggest that averaged SAR wind speeds are a good representation of the wind conditions as experienced by the wind turbines under free stream conditions. Using the wind direction from the SCADA system for the SAR wind retrieval process reduces the RMSE by approximately 0.1 m/s (not shown). This is a small improvement compared to the overall accuracy of the SAR wind retrieval process, thus supporting the SAR processing
choice.

### 3.1.2    Downstream

Figure 3 shows wind speed comparisons for SAR and SCADA downstream and wind directions defined in Table 2. At hub height, the averaged SCADA wind speed is 10.20 m/s and comparisons to SAR give a bias towards higher wind speeds from SAR of 0.64 m/s. The correlation coefficient of 0.97 is good for a linear fit with a slope of 1.06, and the RMSE is 2.12 m/s.
Again, the correlation coefficient and the slope at 10m height are similar whereas the RMSE is lower (1.7 m/s). The mean bias is numerically smaller at 10 m (-0.51 m/s) than at hub height (-0.64 m/s).

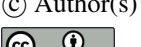



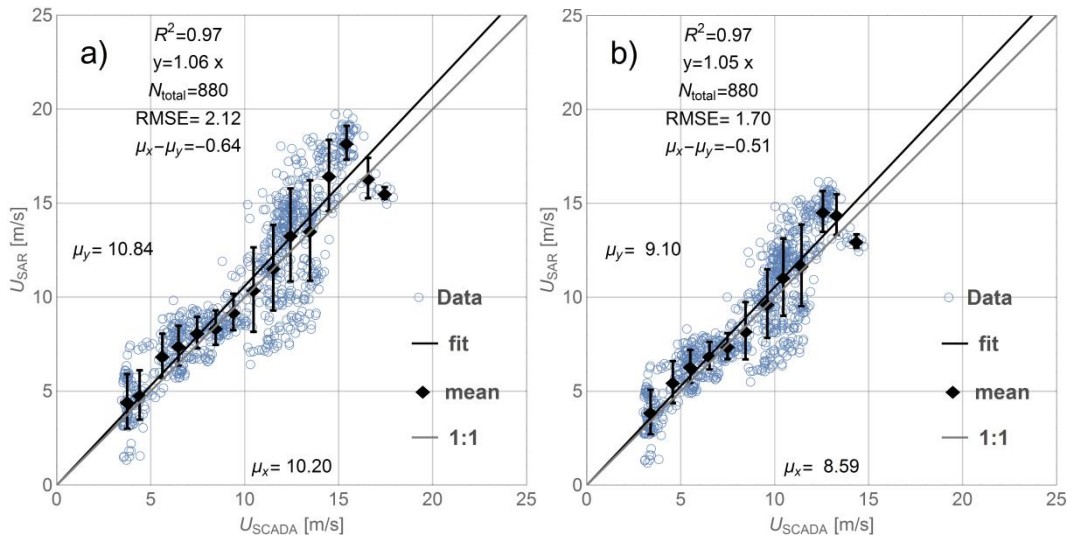

**Figure 3: Comparison between SCADA derived wind speeds ($U_{SCADA}$) and SAR derived wind speed ($U_{SAR}$) downstream of the wind turbines: a) for the turbine hub height (81.6 m), b) for the reference height 10 m.**

The bias is numerically higher downstream than it is upstream of the wind farm. This difference is as expected since the assumption of a logarithmic wind profile may not hold in the wake of the wind farm. The RMSE for downstream conditions is approx. 0.1 m/s lower than for upstream conditions. This result seems counterintuitive, since we expect the assumption of a single logarithmic wind profile from the surface to hub height to be better satisfied upstream than downstream in the wind farm wake. The number of observation pairs is higher upstream

(1026) than for downstream (880), due to coverage of the SAR images and a reduced number of turbine locations downstream for the prevailing westerly wind directions. The different sample size may have an impact on the results for upstream and downstream conditions.

### 3.2    Wind speed variability along Row A

Observations of the past wind conditions are used in wind resource assessment to estimate wind conditions. Satellite SAR

observations are available 10 years before the wind farm at Anholt was constructed. We are investigating how the variability of the mean wind speed at the site could be predicted from SAR winds prior to the wind farm construction. Our analysis of SAR wind maps is complemented by an analysis of numerical simulations from WRF, which are also available prior to the wind farm construction. The overall data availability for SCADA, SAR, and WRF is shown in Table 1 and the number of observations used in this analysis is shown in Table 3.





SAR wind speeds at the turbine locations of Row A are extracted as described in Sect. 3.1 for upstream situations. For the WRF simulations, hourly WRF wind speeds at hub height are interpolated for each of the turbine locations before they are averaged. Both data sets are filtered according to the following conditions: i) Wind directions are between 245° and 275°, which represents a sector where the wind speed variability along the turbine row is expected to be large (Peña et al. 2017); ii)

there is full availability of measurements for all turbine locations along Row A; and iii) wind speeds averaged over row A are above the cut-in wind speed of the wind turbine. The averaged wind speeds are nondimensionalized through division with the respective wind speed at turbine position A15 (see Figure 1b) giving a relative measure of wind speed variability along Row A.

The wind speed variability from SAR and WRF is first examined using two different sampling scenarios for the WRF simulations: the full WRF data set (2002 to 2012) and the WRF samples collocated with the SAR scenes (Fig. 4a). For both scenarios, the WRF simulations show a smooth and monotonically increasing mean wind speed from south to north along Row A. The maximum deviation of mean wind speeds from the two WRF data sets is below 0.5%. This suggests that the reduced sampling rate, which corresponds to the sampling of SAR observations, has little effect on the mean wind speed.

The wind speed variability from SAR observations is less smooth and shows a local maximum at turbine A23. SAR winds are increasing from south to north until they stay approximately constant from turbine A24 on. The wind speed variability from SAR is in good agreement with the two WRF data sets from turbine A01 until A25 where the SAR wind speeds start to decrease.

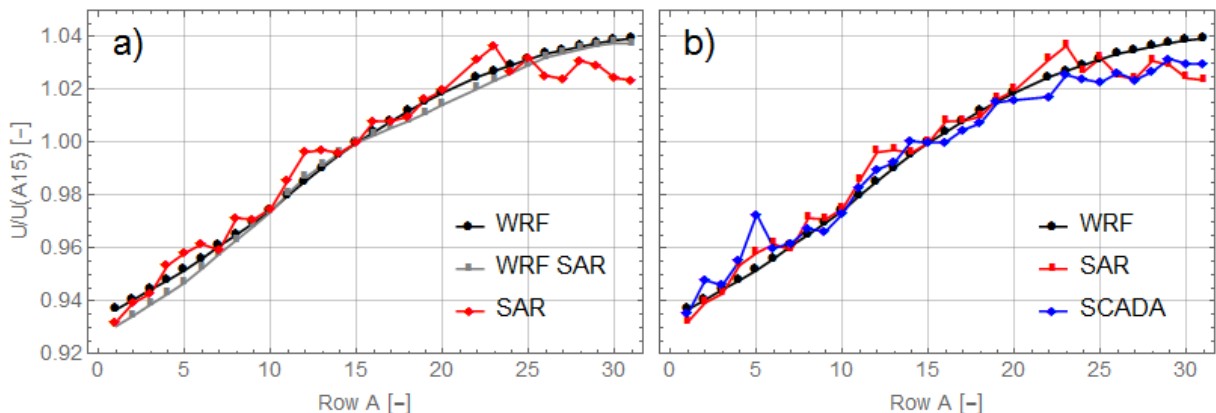

**Figure 4: Average wind speed relative to turbine A15 for wind directions between 245° and 275°. a) Data from WRF (2002-2012) and SAR (2002-2012). The entire time period is used for "WRF" and WRF data coinciding with SAR images are used in "WRF SAR". b) Data from full WRF time series, SAR, and SCADA (2013-2015). No turbine was erected at location A21.**

The relative mean wind speeds from SAR and WRF along Row A are compared to SCADA wind speeds in Figure 4b. All available data from both SAR and WRF are used to best approximate the wind speed climatology from each data set. The

SCADA winds, in contrast, cover a shorter period after the wind farm construction. An unidentified problem at position A05





causes a spike in the wind inversion from the SCADA data but otherwise there is a clear increase of the wind speed from turbine A01 until A20 in agreement with both the SAR and WRF data sets. From position A24, SCADA and SAR winds show a similar behaviour whereas WRF winds are consistently higher and with less spatial variability. We can summarize the findings above as wind speed differences between the Southern-most and Northern-most turbines. The difference $\Delta U_{N,S}$ is defined as:

$$\Delta U_{N,S} = \sum_{i=A28}^{A31} U_i - \sum_{i=A01}^{A03} U_i$$

Where $U_i$ is the mean wind speed at the turbine location. The difference between the Northern and the Southern part of the wind farm is given in Table 3. SCADA and SAR agree within 0.1% while WRF predicts a 1% larger difference than SCADA results suggest.

**Table 3: Sample size and difference between most Northern and Southern turbines $\Delta U_{N,S}$ (three turbine location averaged).**

|  | SAR | WRF SAR | WRF | SCADA |
|---|---|---|---|---|
| Samples N [-] | 72 | 72 | 10524 | 4625 |
| $\Delta U_{N,S}$ [m/s] | 0.92 | 1.02 | 0.98 | 0.95 |
| $\Delta U_{N,S}/U_{15}$[%] | 8.8 | 10.3 | 9.8 | 8.7 |

The wind speed variability along Row A, as shown in Figure 4 and Table 3, is likely caused by varying fetch from the coastline of Djursland. The fetch at different positions along Row A can vary between 16 km and 50 km for the same wind direction, see Figure 1a. The agreement between nondimensional wind speeds from SAR and SCADA is remarkably good. We can conclude that for this site, wind speeds retrieved from SAR imagery could have predicted the relative wind speed gradients well, before construction of the wind farm.

### 3.3 Wind farm wakes from SAR

To investigate the impact of the Anholt wind farm on the wind conditions in the area, we compare wind speeds extracted from SAR wind maps along two transects before and after wind farm construction. With this approach, a baseline of wind conditions before wind farm construction can be determined.

Wind farm wakes in Anholt are analysed for two wind direction sectors. The first sector (75°-105°) represents easterly wind directions and a long fetch. The second sector (255°-285°) represents westerly wind directions and a short fetch, see Figure 1a. Wind direction information from WRF is used as described in Sect. 2.4. SAR wind fields are selected where this wind direction is within one of the two sectors considered here. Three additional criteria are set for SAR wind fields to be included in this analysis: i) the images must fully cover both transects;





ii) the mean wind speed at 10m over the inflow transect is within the interval 3-12 m/s where we expect wind farm wakes to be strongest; and iii) visual inspection does not show any strong signals that are uncorrelated with the wind speed, e.g. rain contamination.

5   Figure 5 shows the position of the two transects. Transect East is located between 2 km and 10 km to the East of the wind farm and transect West is located between 4 km and 6 km to the West of the wind farm. Along each of the transects, wind speeds are extracted from all available SAR wind maps and averaged over rectangular bins of 1 km (in transect direction) and 1.5 km (perpendicular to transect direction). Resolution cells with more than 5 m/s difference from the median within each bin are filtered out as they likely result from reflection from ships.

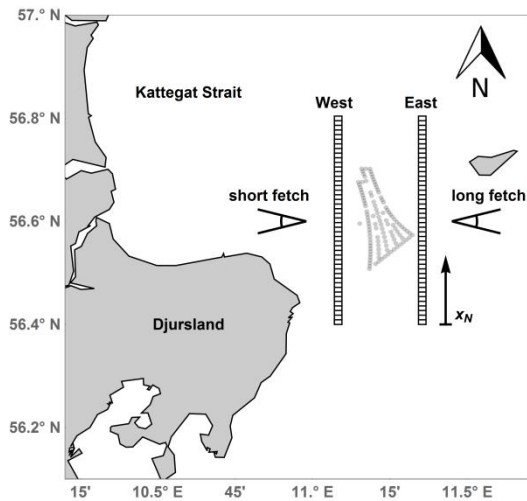

**Figure 5: Location of the Anholt wind farm and investigated transects. Two transects "West" and "East" are following the North/South direction.**

Wind speed pairs extracted at the same latitude from the East and West transects are assumed to be upstream or downstream of each other for the two directional sectors investigated here. We can calculate the difference $\Delta u_i$ between upstream and

15   downstream observations depending on $x_N$:

$$\Delta u_i(x_N) = u_{i,up}(x_N) - u_{i,down}(x_N)$$

where $u_{i,up}(x_N)$ and $u_{i,down}(x_N)$ are the wind speeds on transect "upstream" and "downstream" respectively. From $\Delta u(x_N)$ we can calculate the mean difference $\Delta U(x_N)$ and the standard error $SE(x_N)$. As defined here, a positive $\Delta U$ corresponds to a wind speed reduction on the downstream transect.



### 3.3.1 Long fetch

For situations with easterly winds the transect "East" is upstream and transect "West" downstream of the wind farm. The fetch is approximately 80 km to the East with exception of the Anholt Island, see Figure 1a. A total of 49 SAR wind maps live up to our selection criteria. Of these, 35 were acquired by Envisat before the wind farm was constructed and 14 were

acquired by Sentinel-1 after the wind farm construction. Figure 6a shows the average wind speeds along upstream and downstream transects before the wind farm construction. The wind speeds at the same latitude are very similar over the distance 0 km to 32 km. This is as expected since there is open water between the transects and the fetch is long. At 32-37 km where Anholt island is upstream of both transects, the wind speeds on the upstream transect are slightly lower compared to those along the downstream transect. This is likely caused by the lee effects from the island.

Figure 6b shows the average wind speed along the two transects after the wind farm was constructed. The wind speed along the downstream transect shows a reduction between 11 km and 30 km. The wind speed along the upstream transect remains between 7.3 and 7.6 m/s from 0 km to 25 km and decreases further North. The number of observations is much lower than before wind farm construction.

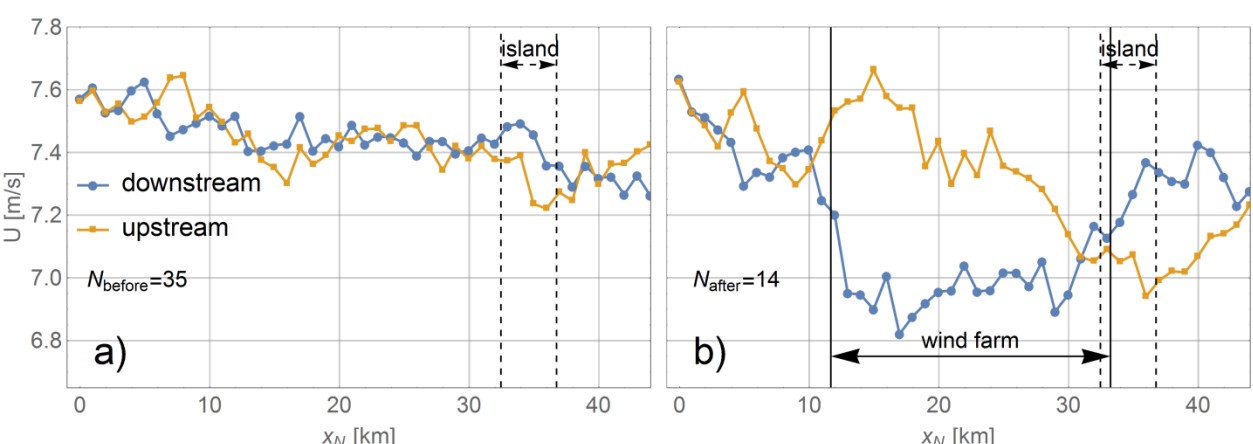

**Figure 6: Wind speed transects from a) before and b) after wind farm construction for wind direction between 75° and 105°. East is upstream and West is downstream of the (potential) wind farm location. The position of the wind farm to the East/West and Anholt Island to the East of the transects are indicated.**

Figure 7 shows the mean wind speed differences $\Delta U$ with on standard error $SE$ indicated by the shaded areas. The average density of turbines between the upstream and downstream transects are shown at the top. Before wind farm construction the

differences range from -0.2 m/s to 0.2 m/s from 0 km until 30 km. $\Delta U$ is negative from 29 km until 37 km around the position of Anholt Island, likely corresponding to a lee effect of the island. After wind farm construction the influence of the wind farm is clearly visible from a difference of 0.3 m/s to 0.75 m/s between 11 km and 27 km. This coincides with the distance where the highest density of turbines is found. Ranges of the standard error are also clearly separated. Around the location of Anholt island, the differences are slightly negative and similar to the differences found before the wind farm

construction. At 6 km, a peak around 0.3 m/s appears. The reason for this peak is unclear but could be non-wind effects such




as bathymetry-current interaction or remaining effects of hard targets which influence the radar backscatter and thus the wind speed retrieval.

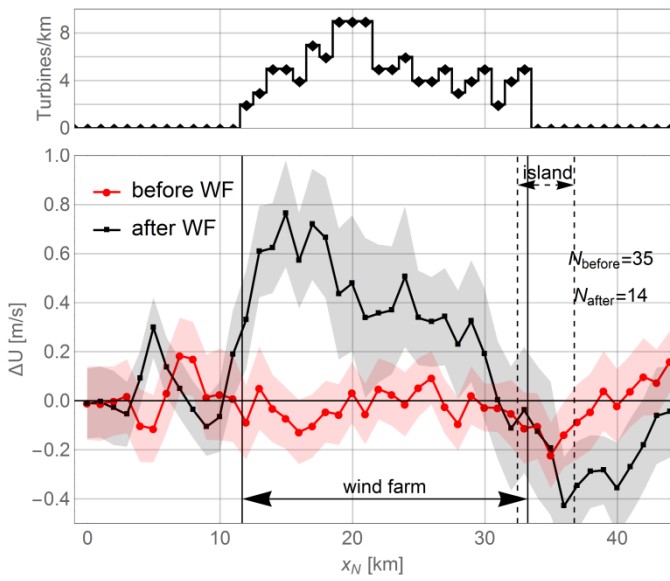

**Figure 7:** Top: Density of turbines per unit kilometre between the transects. Bottom: Mean difference between wind speeds on the upstream and downstream transect before and after construction of Anholt wind farm. Vertical lines indicate the position of the wind farm and dashed lines the position of the island Anholt to the East. The shaded area represents one standard error around the mean.

A sample size of 35 images creates the baseline of the wind conditions before construction of the wind farm. SAR wind speeds after construction show a clear wake, both absolute and relative to the state before construction of the wind farm, see Figure 6 and Figure 7. Even though the sample size of 14 images after wind farm construction is small, the indication of the wind farm wake is strong.

### 3.3.2 Short fetch

For situations with westerly winds the transect West is located upstream and transect East is downstream of the wind farm. The fetch is between 16 and 50 km to the West, see Figure 1a. Average wind speeds along the two transects are analysed in a similar manner as described for long fetch situations in Sect. 3.3.1. A total of 92 images before and 31 after wind farm construction fulfil the selection criteria. Figure 8 shows the averaged wind speeds. The wind speeds are increasing from south to north along both transects. Wind speeds from before wind farm construction in Figure 8a are consistently lower for the upstream compared to the downstream transect. This is expected due to the increasing wind speed further offshore. All transects in Figure 8 show lower wind speeds in the Southern end than in the Northern end. This variability in the wind speed is similar to the one found in Sect. 3.2 and likely caused by the variation in fetch along the transects. Wind speed differences and standard error are calculated similar to Sect. 3.3.1 and are shown in Figure 9.





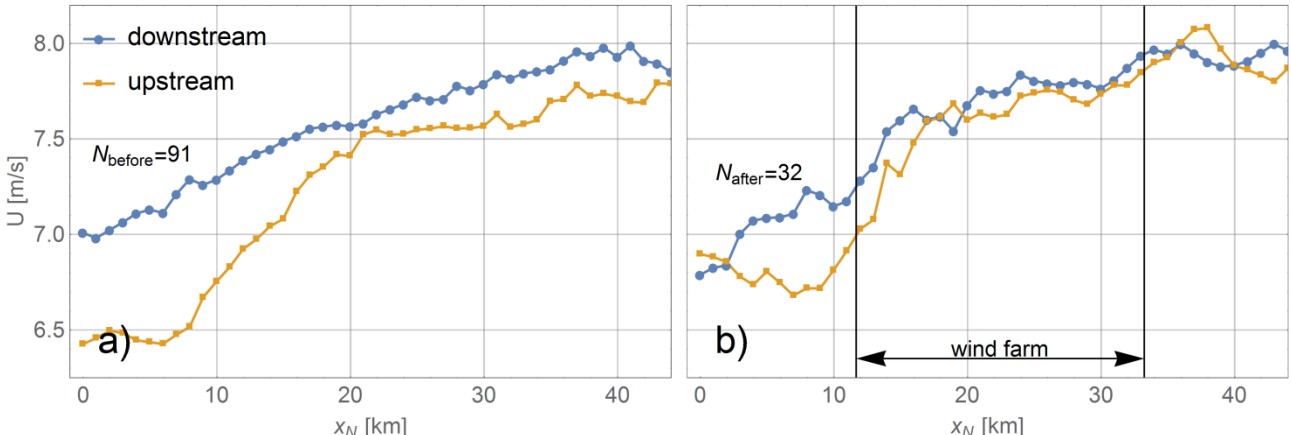

**Figure 8: Wind speed transects similar Figure 6 from a) before and b) after construction for wind direction between 255° and 285°. Transect West is upstream and East is downstream of the (potential) wind farm location.**

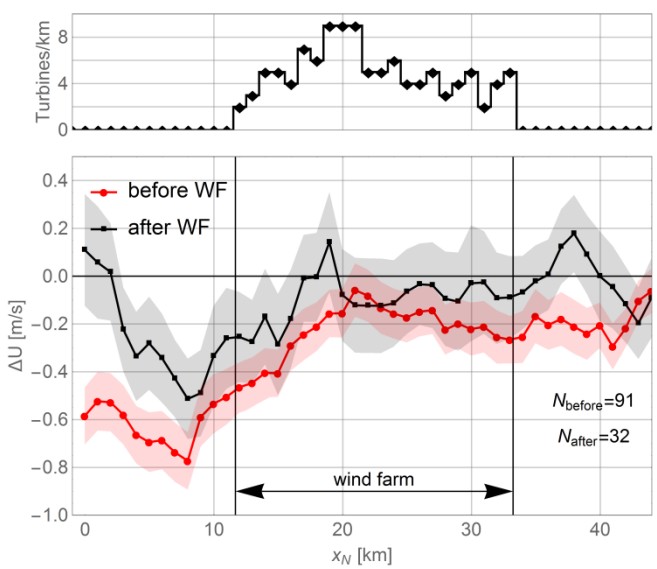

**Figure 9: Wind speed difference similar to Figure 7 for short fetch situations with wind directions between 255° and 285°.**

The wind speed difference before wind farm construction ranges between -0.7 m/s and -0.4 m/s for the area South of the potential wind farm. Further North from 17 km on the difference ranges between -0.3 m/s and -0.1 m/s. This is consistent with a short fetch in the south where wind speed is expected to speed up more between the transects than in the northern part

10    with longer fetches. Wind speed differences after construction of the wind farm show roughly the same pattern except between 0 km and 8 km where differences are large. No clear evidence of wind farm wake effects are found since there is no significant difference is noted between the average wind speeds before and after wind farm construction.The effect of a wind



farm wake on the SAR wind fields is likely too weak to be detected compared to the strong wind speed gradients. The number of observations before wind farm construction is approximately three times larger than after. The averaged wind speed after construction is less smooth. The convergence to a smoother mean wind speed is expected in the future with more observations from Sentinel-1 A and B become available.

## 3.4 Wind farm wakes and gradients

To analyse the cumulative effect of coastal wind speed gradients and the wind farm wake effect, four parallel transects are defined perpendicular to the coastline following the orientation of row 1. Figure 10 shows a reference transect to the north of the Anholt wind farm (a) and three transects across the wind farm (b, c, and d). Average wind speeds are extracted along these transects similarly to Sect. 3.3.

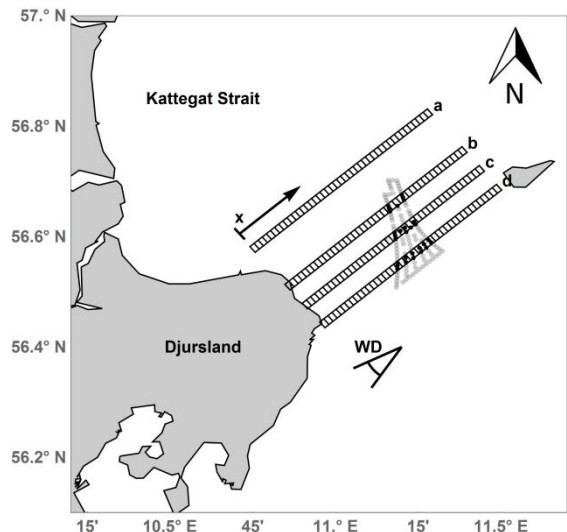

**Figure 10: Transects used for analysis of wind farm wakes and coastal gradients. Turbines inside the transects are marked in black. Origin and direction of coordinate x, and the wind direction range (WD) used for the selection of satellite scenes are indicated.**

For this analysis, SAR wind maps are selected according to the following three criteria: i) there is full coverage over all four transects, ii) SAR wind speeds at 10 m upstream of the wind farm are between 3 m/s and 12 m/s, and iii) the wind is coming from directions within the sector 214.5°-244.5° centred around the transect orientation and roughly corresponding to prevailing wind direction at the site. WRF outputs are used to determine the wind direction as described in Sect. 2.4. A total of 57 images before and 35 after the wind farm construction fulfil these criteria.

SCADA wind speeds are extracted for the wind turbine locations covered by transect b, c, and d. The following criteria are used for filtering of the SCADA wind speeds: i) the turbine locations are within the transects and data is available for all those turbines (cf. Figure 10), ii) the SCADA wind direction ranges between 214.5° and 244.5°. A total of 3371 10-minute mean values of SCADA wind speeds live up to these criteria. Data from SAR and SCADA are not collocated in time. The




wind turbines are placed in rows oriented from North to South. SCADA wind speeds are averaged for each row segment within each transect.

SAR wind speeds are presented as differences with respect to a reference wind speed, $U_{ref}$ upstream of the wind farm (for transect b, c, and d). For transect a, the reference point is at the same x position as for transect b. SCADA winds are shown
5   as wind speed differences compared to the free stream turbines in row A. Wind speed differences along transects a to d are shown in Fig. 11. Before the wind farm construction, there is a clear coastal wind speed gradient with increasing wind speeds with distance from the coastline for all four transects. For the reference transect a, the deviation between the results before and after wind farm construction is below 0.2 m/s.

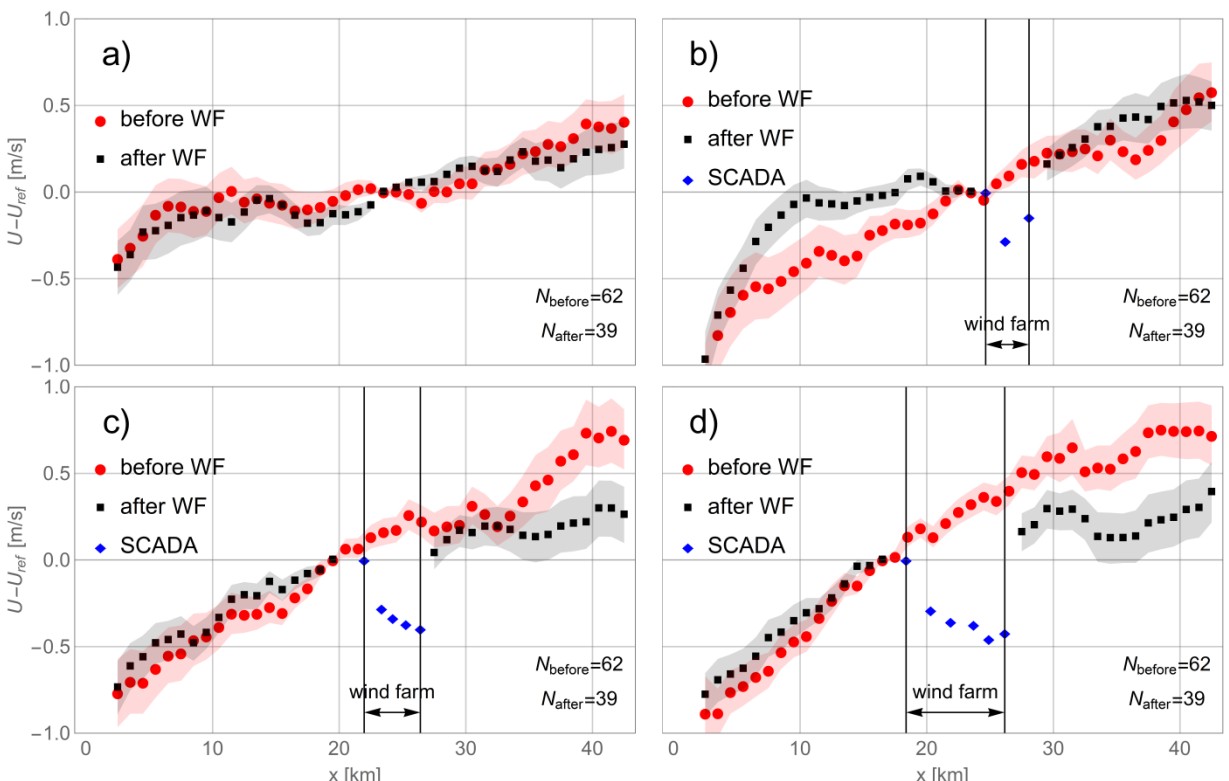

10   **Figure 11: Wind speed differences from SAR along transects a to d before and after construction of the wind farm. Differences calculated from SCADA wind speeds are also shown and the position of the wind farm is indicated.**

Wind speeds upstream of the wind farm (transect b, c, and d) clearly show wind speed gradients, both before and after wind farm construction. For transect c and d wind speeds differences before and after the wind farm construction agree within 0.2
15   m/s, but large deviations are found at transect b. The fetch along transect b is very sensitive to the wind direction for South-





westerly winds. Differences in the distribution of wind directions between SAR data before and after wind farm construction could be the reason for the large deviations.

Wind speeds downstream of the wind farm show a positive wind speed gradient along for transects b, c, and d. The wind speed on transect b is similar before and after wind farm construction. This transect crosses a narrow part of the wind farm

with only three turbine rows. Transects c and d cross a larger number of turbines and show a significant change of the wind speed after the wind farm construction. We attribute this change to wake effects of the wind turbines.

SAR wind speeds cannot be retrieved correctly within the wind farm itself due to radar reflection from the turbines. The SCADA wind speeds for turbines within transect b to d are used instead to describe the wind speed behaviour within the wind farm. The SCADA wind speeds suggest a reduction of wind speeds downstream of turbine row A which is most

pronounced for transect c and d that cross many turbine rows.

SCADA wind speeds show the wind farm wake a as reduction in wind speed compared to the upstream turbine. SAR winds on transect c and d show a reduction of wind speed compared to the situation before construction of the wind farm. The deviations between these two types of wind speed information are between 0.3-0.6 m/s. Differences between SAR and

SCADA winds can be attributed e.g. to: difference in the location with SCADA winds at the turbine positions and SAR winds downstream of the wind farm, differences in the sample size and measurement that are not collocated in time, or differences in the vertical position of the measurements. SCADA data are derived at the turbine operating height whereas the SAR wind retrievals are based on observations of the sea surface. The strongest wind turbine wake effect is expected at the turbine hub height, which is consistent with a stronger wake from SCADA winds compared to SAR.

**4    Discussion**

We have demonstrated how an extensive archive of SAR wind maps can be used to identify the combined effects of a complex coastal geometry and wind farm wakes on the mean wind conditions around the Anholt wind farm. Our results illustrate how wind maps retrieved from SAR can predict the wind conditions that offshore wind turbines and whole wind farms experience before a wind farm is constructed.


For the first time, wind speeds derived from the SCADA system of an entire wind farm have been compared to SAR wind speeds, see Figure 2. The correlation for free stream conditions is good and the slope of the fit is very close to one. This result is encouraging for using SAR derived mean wind speeds to predict wind conditions as experienced by the wind turbines. GMFs used for SAR wind retrieval are tuned using observational data from buoys in the open ocean. Influences of

internal boundary layers caused by the roughness change between land and sea, or effects of limited fetch on the ocean surface roughness are not fully accounted for. These effects are hard to quantify, but the RMSE compared to lidar



measurements in the coastal zone is between 1.3 and 1.4 m/s (Ahsbahs et al. 2017). The SAR wind speed retrieval process needs a wind direction as an input. Readily available SAR wind maps using a global model wind direction are used throughout this study. Therefore, uncertainties in the modelled wind direction translate into errors in the wind speed retrievals.

The wind retrieval process assumes a logarithmic wind profile. Influences of atmospheric stability on the instantaneous comparison between SAR and SCADA wind speeds cannot be accounted for without site specific measurements. To overcome this problem, SAR wind speeds can be presented relative to a reference location as shown in our analyses. Assuming Monin Obukov theory, constant stability, and roughness over the domain introduces a stability correction factor

that is independent of the location and height. The relative wind speed is thus independent of height. These assumptions will not be valid at all times, but the extrapolation error of the mean wind speed from 10m to hub height is expected to decrease when the mean wind speed is divided by a reference location.

The atmospheric boundary layer changes significantly with the presence of wind farms. This will affect our comparisons of

SAR and SCADA wind speeds downstream of the wind farm, see Figure 3. The correlation is good but the bias towards higher wind speeds from SAR has increased compared to the analysis upstream of the wind farm. A logarithmic wind profile is no longer valid and the shear close the ocean surface increases. The largest wake deficit is located at hub height (Porté-Agel et al. 2011). This could cause an overprediction of the SAR wind speed when extrapolated. Additionally, SAR winds are retrieved between 600m and 2600m downstream of the turbine position but are compared to SCADA wind speeds at the

turbine location and the wake is likely to recover. This is also consistent with the difference between SAR and SCADA winds in Figure 11. To better quantify wind farm wakes from SAR images, further work is needed to understand how wakes interact with the ocean surface and how this influences SAR wind retrievals.

The Anholt wind farm is experiencing strong variability in the wind speed along the westernmost row (Row A) for the

prevailing wind directions from 245-275°. The normalized mean wind speed obtained from 72 SAR images before construction of the wind farm agrees very well with results from SCADA winds of the first 2.5 years of wind farm operation. The mean wind speed between South and North of row A increases by 8.7% in the SCADA wind speeds and 8.8% in SAR derived wind speeds, see Table 3. SAR wind maps are valuable for characterization of large scale flow phenomena such as wind speed variations over long rows of turbines.


Nondimensional wind speeds from WRF at the turbine locations also predict wind speed variability very similar to results from SAR and SCADA. Models such as WRF are powerful tools to identify good wind resources, but cannot fully replace observations of the wind conditions on site. The presented analysis of SAR wind maps can complement modelling efforts by introducing an independent measurement for comparison, since both data sets are available before construction of a potential



wind farm. The good agreement between WRF and SAR with regard to wind speed variability adds confidence to assessments of the wind resource.

Anholt wind farm has irregular turbine spacing and the shape is elongated. Methods applied at other offshore wind farm sites
for analysing wakes in SAR wind maps are less suitable for Anholt (Hasager, Vincent, et al. 2015). A new approach for analysing wind farm wakes from SAR images has therefore been suggested, which explores the difference of SAR wind maps before and after the wind farm was constructed. The wind farm wake effects are analysed along transects approx. perpendicular to the wind direction on the upstream versus the downstream side of the wind farm and along transects crossing the wind farm aligned with the wind direction.

For situations with a long fetch, perpendicular transects before wind farm construction provide a suitable baseline to check averaged differences between upstream and downstream transects, see Figure 7. The wind farm wake measured from SAR shows a structure that roughly follows the turbine density of the wind farm. For fetch limited wind directions with the presence of a complex shore line, this approach does not show a wind farm wake, see Figure 8. Here transects crossing the
wind farm can be used instead to investigate how the coastal wind speed gradient and wakes of the wind farm interact, see Figure 11. No wind speed reductions compared to the upstream reference point are found. Two transects going through an area of high wind turbine density show a reduction of wind speed between after wind farm construction compared to before. The complexity of the shore line makes results very sensitive to the wind direction. This is likely the cause that no sign of a wake is found for transect b in Figure 11. Wake analyses for non-regular shaped wind farms are possible, but a strong coastal
wind speed gradient can easily dominate over the wind farm wake effects. Further studies at locations with simple geometry of the coastline would help to understand the interplay of wind farm wakes and coastal wind speed gradients.

SAR wind maps are suitable for analysing large scale wind conditions and they can show the combined effects of different flow phenomena. In this analysis wind farm wakes, coastal wind speed gradients and wind speed variability from differing
fetch occur simultaneously. It is challenging to identify the contribution of one particular flow phenomenon, e.g. wind farm wakes from this data. In contrast to engineering wake models as FUGA or Park that are run with a single wind speed and direction, SAR wind maps capture the full picture of the flow around a wind farm. The presented methods can easily be repeated for any potential offshore wind farm site.

The presented SAR data archive goes back to 2002 and offers the possibility of reference measurements before most of the current offshore wind farms were constructed. The analyses presented in this study will gain confidence as the satellite data archives are growing over time. With Sentinel-1 A and B, two new satellites are acquiring new scenes on a daily basis which are available in the public domain. This makes SAR wind maps more accessible and the time is right to develop tools that are tailored to the needs of the offshore wind industry.





## 5    Conclusion

Large archives of SAR wind maps have recently become publically available and the sampling frequency of the
measurements has increased significantly with the European SAR missions Sentinel-1 A/B. Readily available SAR based
wind speed maps represent a computationally and monetarily cheap source of information about the large scale wind speed
variability offshore. The maps are available in hindcast and may thus be used from the earliest stages of a wind farm project.
We have demonstrated that wind speed maps retrieved from SAR observations of radar backscatter can be used to predict the
spatial wind speed variability at a potential wind farm site before construction begins. The satellite based wind speed maps
can also be used for characterization of wake effects around existing wind farms and to partially determine the cumulative
effects of coastal wind speed gradients and wake effects.

Wind speeds retrieved from SAR correlate well with the SCADA derived wind speeds for the turbines at Anholt wind farm.
RMSEs are 2.23 m/s and 2.12 m/s for comparisons upstream and downstream of the wind farm, respectively. Wind farm
wakes are detected from SAR wind fields using a long time series with measurements before and after construction of the
wind farm. This approach is powerful, since a baseline of wind conditions before the construction is available. Strong
indications of wind farm wake effects are found for wind directions leading to a long fetch with a maximum deficit of
0.7m/s. Wind farm wakes at fetch limited conditions are harder to identify due to the complex interplay of different effects
such as varying fetch and coastal wind speed gradients on the mean wind speed.

Our results indicate that SAR wind maps can resolve smaller-scale wind variability, which is seen from the SCADA wind
speeds but might not be present in the WRF models. In the early stages of planning an offshore wind farm, the wind speed
variability given by SAR wind maps may help in the planning of on-site measurement campaigns. Alongside with model
simulations, satellite based wind maps represent a valuable resource to introduce large scale on-site measurements early in
an offshore wind farm project.

**Data availability:**

SAR wind fields are available at https://satwinds.windenergy.dtu.dk/ and WRF model runs can be made available upon
request. SCADA data is not available for publication.

**Author contribution:**

Tobias Ahsbahs developed methods and code. Merete Badger and Charlotte B. Hasager provided the processed SAR wind
maps and contributed with guidance and comments. Kurt S. Hansen prepared the SCADA data and Patrick Volker provided
the WRF data. Tobias Ahsbahs prepared the manuscript with contributions from all co-authors. This work is part of Tobias
Ahsbahs' PhD under supervision of Merete Badger.

**Competing interests:**





The authors declare that they have no competing interest.

**Acknowledgements:**

We would like to acknowledge Ørsted for granting access to data from the Anholt wind farm, Johns Hopkins University Applied Physics Laboratory and the National Atmospheric and Oceanographic Administration (NOAA) for the use of the

SAROPS system, and ESA for providing public access to data from Sentinel-1A. Personal thanks to Nicolai G. Nygaard from Ørsted for his approval and comments.

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
