# Peer review of "Applications of satellite winds for the offshore wind farm site Anholt"

_Wind Energy Science, 2018_

## Referee Comment (RC1) · Anonymous Referee #1 · 5 Mar 2018

The manuscript displays data from different sources (SCADA, SAR, WRF) in order to analyse aspects of the wind field within and around the Anholt wind farm. The main purpose is to analyse the reliability of SAR data for this purpose. This is an important topic since data sources offshore are limited and SAR data seem to be promising. Nevertheless, the manuscript in its present stage is not fully convincing. I suggest major changes before a publication can be recommended.

Points for revision:

Major issues:

(1) The main point of this manuscript is the comparison between SCADA and SAR data and the ability of the SAR data to trace the wake of the wind farm. The manuscript

should concentrate on this. Therefore, I suggest to skip the comparison with the WRF data. The WRF data are mentioned only marginally in the Abstract and in the Conclusions of the manuscript. This illustrates their minor importance for this investigation.

(2) There is considerable difference in data quality between SCADA and SAR data. First of all, SAR wind data at 10 m height is not a measurement but a product which arises from the stipulation of severe assumptions involving GMFs. That this could be a source of uncertainty is not even mentioned in the manuscript. Then, the 10 m SAR data must be extrapolated to the hub height of 81.6 m. This is done here by assuming a logarithmic wind profile. But the authors do not know how well this logarithmic profile depicts the reality. We know from earlier studies that the wind regime in the Baltic (Smedman and colleagues in the 1990s) is very often characterised by warmer air over colder waters. This leads to the formation of internal boundary layers, low-level jets and non-logarithmic vertical wind profiles. This problem must be addressed in a manuscripts which intends to promote the use of SAR data in offshore wind energy especially in the Baltic.

Further issues:

(3) The first paragraph of the Introduction summarises the impact of inhomogeneous coastal areas on offshore wind fields. Here the work of Dörenkämper et al. should be mentioned (Dörenkämper, M., Optis, M., Monahan, A., & Steinfeld, G. (2015). On the offshore advection of boundary-layer structures and the influence on offshore wind conditions. Boundary-Layer Meteorology, 155(3), 459-482).

(4) page 2, lines 22-27: aircraft measurements at hub height in the far wake of large offshore wind farms are now available. See: Platis, A., S.K. Siedersleben, J. Bange, A. Lampert, K. Bärfuss, R. Hankers, B. Cañadillas, R. Foreman, J. Schulz-Stellenfleth, B. Djath, T. Neumann, S. Emeis, 2018: First in situ evidence of wakes in the far field behind offshore wind farms. Scientific Reports, 8, 2163. DOI: 10.1038/s41598-018-20389-y

(5) page 2, line 2: a reference is missing

(6) page 2, line 8: wind farm parameterisation in WRF: here, the parameterisation available in WRF by Fitch should be mentioned: Fitch, A. C., Olson, J. B., Lundquist, J. K., Dudhia, J., Gupta, A. K., Michalakes, J., & Barstad, I. (2012). Local and mesoscale impacts of wind farms as parameterized in a mesoscale NWP model. Monthly Weather Review, 140(9), 3017-3038.

(7) How has the correlation coefficient displayed in Figure 2 been computed? From the bin means (black diamonds in the Figure) or from the full data set (grey circles in the Figure)? It actually does not look like that the grey circles can be explained by such a high correlation coefficient.

(8) What is displayed in the last line of Table 3?

(9) Subchapter 3.3.2: it does not really become clear why there are no discernible wake effects in the "short fetch" case.

(10) Chapter 3.4 and Figure 11 (related to the preceding comment): Figs. 11 c and d are in contradiction to Figs. 8 b and 9. While there are clear wake effects in Figs. 11 c and d, there are none in Figs. 8 b and 9. The wind direction in both groups of figure is nearly identical. This contradiction should at least be addressed in the manuscript. This contradiction seems to be a clear hint that SAR data is not easily interpretable (see the above comment no. 2).

---

## Referee Comment (RC2) · Anonymous Referee #2 · 7 Mar 2018

General Comments:

This paper addresses relevant and interesting science questions. The comparison of SAR and SCADA is novel . It is generally well written and structured so as to be easy to follow. The methodology is clearly presented, but missing some important details. Wind speeds retrieved from SAR correlate well with the SCADA derived wind speeds (P21,L11) and this is a useful result. The results also show the SAR derived wind speeds reproducing the measured wind speed variability along turbine row A well. However, errors and uncertainties arising from the method are not fully explored and considered. I don't feel that the results presented fully support the conclusions drawn and so this paper should be revised before final publication.

Specific Comments:

Comments regarding the methodology:

P4, from L16: Why select this GMF, these reanalysis data sets for derivation of SAR wind speed? Is this essentially describing how the DTU archive of wind speed maps is derived? If so, it would be clearer to turn this around to say that the archive of processed wind maps is used, and this archive is derived using SAROPS, CMOD5.N etc.

P5, L23 : It is not clear what "averaging (WRF) wind direction at the same locations as for SCADA derived wind direction" means. Is this averaging across some time window or spatially, and to what end?

P6,Table 1: I am quite unclear on what 'Wind Direction' refers to. Isn't SAR wind speed using reanalysis data input all the time? Is this the wind direction used to select scenes, eg to determine that the wind direction is long or short fetch?

P6, L9 : Why the hexagonal shape? How does it relate to SAR wind speed resolution cells? P10, L4 : Why is variation expected to be large? (This is kind of explained P11 L 14-15, but why not hypothesised here? It would be clearer.)

Comments regarding treatment of Errors and Uncertainties:

P2,L10 : It is mentioned that the absolute accuracy of SAR derived wind speeds are low compared to mast measurements. This needs further elaboration and consideration in order to draw conclusions from the results.

P8, L9 and P19, L3: Wind direction input is picked out as a source of error in deriving SAR wind speeds, but is not explored. There is a passing reference to having also looked at SAR winds derived with SCADA wind direction as input. This would seem to offer an opportunity to explore and possibly quantify to some degree the uncertainties arising from wind direction input.

P5,L30: For SAR based wake studies it is assumed that turbines are operational. With SCADA data available in parallel with SAR, the uncertainties arising from this assumption could be quantified allowing you to draw firmer conclusions from the results found.

P5, L17; P7,L12; and P19, L9 and 17 : The simplification of adjusting between heights using a logarithmic wind profile is highlighted as a likely source of error. Please expand on this. What might be a better option, if you had information about the stability conditions? Uncertainties arising from this could be tested by trying other assumed atmospheric conditions and height adjustments and seeing to what degree they affect the delta U results. The discussion on p19 seems to contradict itself, suggesting that the height correction method should not be affected between upstream and downstream measurements as stability conditions will be similar for both (L9) but that downstream the logarithmic profile is no longer expected to be valid (L17).

P10,L25 : An anomalous result at position A05 is dismissed as a problem and not real, but can this be justified. Can the SCADA data reveal evidence for this assertion?

Comments regarding the conclusions drawn from the results:

P9,L11 : I am not satisfied by the suggestion that the lower RMSE seen downstream might be due to sample size. Lower RMSE would seem to imply that the SAR 'model ' data better fits the SCADA observations downstream.

P21, L14: The wakes study with SAR does appear to show a wind farm wake effect in the long fetch scenario, but I am not satisfied that this allows the conclusion that 'strong indications of wind farm wake effects. . .' . This is because the short fetch scenario shows no, or weak effects, and in the long fetch scenario there appears to be some unexplained wind speed up after the wind farm is constructed in the region north of the island (P14,figure 7). I think it can only be concluded that it's a promising technique and should be revisited in the future with more post wind farm SAR data.

P17,L15 : The explanation for the deviations between before and after upstream wind speeds in transect b is not at all satisfactory. There is no explanation of why transect b might be more sensitive to wind direction than say d. This theory should be further

explored by considering any bias in the wind direction distribution between the before and after wind farm images and comparing this to the transect locations relative to the coast.

Technical Corrections:

P1, L20 : Give the date that these wind capacity numbers are valid for.

P2, L14,18 : Insert brackets around these references (name and year) for consistency with others in the paper.

P2 , L6 : "wind is causing" should be "wind causes".

P3, L2 : Missing reference

P4, L6 : Suggest inserting ", and was " between 400MW and constructed to make the sentence clearer. P4, L16: Insert "," after wind speed retrieval.

P5, L5: Explain "curtailed" for the benefit of readers from the remote sensing rather than energy community.

P5,L8 : "on at edge" should be "on the edge" or "at the edge".

P11,L6 : Number the equation and then reference it in Table 3.

P13, L18 : "with on standard error" should be "with one standard error".

P16, Figure 10 : The turbines marked in black are not clear to me.

P18, L11 : "a as" should be "as a".

P19, L24: "is experiencing" should be "experiences".

P20: L7: "approx." should be "approximately".

---

## Author Comment (AC2) · 17 Apr 2018

Dear Referee,

thank you for your constructive comments and technical corrections. Please find detailed answers to your comments in the supplement along with an updated version of the paper.

Best regards, Tobias Ahsbahs et.al.

Please also note the supplement to this comment:
https://www.wind-energ-sci-discuss.net/wes-2018-2/wes-2018-2-AC2-supplement.pdf

---

## Author Response (AR1)

**Answer to anonymous Referee #1**

Dear Referee #1,

We would like to thank you for the suggested revisions to our manuscript. The issues you raised were constructive and helpful to improve the quality of this paper and we have carefully considered them and revised this manuscript. Find enclosed our detailed answers to the comments that you raised.

Please note that the numbering of the Figures have changed since we added two new Figures. In our answers we refer to the updated Figure numbers and give the Figure numbers for the earlier version in brackets. Textual changes where we have considered your comments in the manuscript are given by page and line number for the new manuscript (not the one with track changes) and in brackets for the initial manuscript. Additionally, we have made a few changes where we felt the text could be improved and clarified.

Best regards,

Tobias Ahsbahs et. al.

**Points for revision:**
**Major issues:**

*(1) The main point of this manuscript is the comparison between SCADA and SAR data and the ability of the SAR data to trace the wake of the wind farm. The manuscript should concentrate on this. Therefore, I suggest to skip the comparison with the WRF data. The WRF data are mentioned only marginally in the Abstract and in the Conclusions of the manuscript. This illustrates their minor importance for this investigation.*

Answer: We find the results from WRF in Figure 6 (4) relevant to interpret the results obtained from SAR. We agree with you, that the results are not particularly present in the Abstract or the Conclusion but would like to highlight their importance there instead of omitting the result.
We find WRF modelling results are important here for two reasons: 1) 72 SAR images are the basis for calculations of non-dimensional mean wind speeds for row A. WRF results for the full time series and the coinciding time stamps show very similar results. We find it important to show the effect of downsampling in the model domain to support the selection of SAR images. 2) Both SAR and WRF (or other NWP models) are available before construction of a wind farm. In this case they agree well and we would argue that an agreement of two different data sets adds confidence to their accuracy. We would like to suggest to explore more cases in a future study to determine conditions where disagreement is found in wind speed variability over coastal waters.

Changes:
P22 L5 to L10 (P19,L25 to L26)
P22 L17 to L24 (P20,L2)
P24 L25 to L28 (P21,L19)

*(2) There is considerable difference in data quality between SCADA and SAR data. First of all, SAR wind data at 10 m height is not a measurement but a product which arises from the stipulation of severe assumptions involving GMFs. That this could be a source of uncertainty is not even mentioned in the manuscript. Then, the 10 m SAR data must be extrapolated to the hub height of 81.6 m. This is done here by assuming a logarithmic wind profile. But the authors do not know how well this logarithmic profile depicts the reality. We know from earlier studies that the wind regime in the Baltic (Smedman and colleagues in the 1990s) is very often characterised by warmer air*

*over colder waters. This leads to the formation of internal boundary layers, low-level jets and non-logarithmic vertical wind profiles. This problem must be addressed in a manuscripts which intends to promote the use of SAR data in offshore wind energy especially in the Baltic.*

Answer: Regarding general assumptions of the GMF to retrieve wind speed: This is a good point. We modified the manuscript in Sect. 2.2 to reflect more on the assumptions and uncertainties of SAR wind fields.

Changes:
P5 L1 to L9 (P4,L14)

Regarding stability correction, we added simulation results from WRF for the wind farm site to give indication about the atmospheric stratification (see Fig. 1) and moved the introduction of wind speed extrapolation into a separate subsection under Sect. 2.5. Using modelled stability results for extrapolation of SAR wind fields does not increase the accuracy compared with in situ measurement (Badger, M. et al., 2016. Extrapolating Satellite Winds to Turbine Operating Heights. *Journal of Applied Meteorology and Climatology*, 55(4), pp.975–991.). A simple stability correction assuming averaged profile to be near stable and near unstable has been implemented. These results shall illustrate the impact of stability of the upstream comparisons in Fig.4 (2) where the bias changes but the RMSE changes marginally. We have added a similar test for results presented in Fig. 6 (4) where the results are almost unaffected due to the presentation as a wind speed ratio. Additionally, we have raised this issue in the Discussion.

Changes:
Add Sect. 2.5 P6 L24 to P7 L10
P9 L13 to L17 (P7 L8 to L13)
P10 L8 to L9 (P8,L10)
P13 L16 to L22 (P11 L 16)
P22 L12 to L15 (omitted P19,L6 to L12)

**Further issues:**
*(3) The first paragraph of the Introduction summarises the impact of inhomogeneous coastal areas on offshore wind fields. Here the work of Dörenkämper et al. should be mentioned (Dörenkämper, M., Optis, M., Monahan, A., & Steinfeld, G. (2015). On the offshore advection of boundary-layer structures and the influence on offshore wind conditions. Boundary-Layer Meteorology, 155(3), 459-482).*

Answer: This is an interesting and very relevant study and we have added it to the Introduction.
Changes: P1,L27 (P1,L27):

*(4) page 2, lines 22-27: aircraft measurements at hub height in the far wake of large offshore wind farms are now available. See: Platis, A., S.K. Siedersleben, J. Bange, A. Lampert, K. Bärfuss, R. Hankers, B. Cañadillas, R. Foreman, J. Schulz-Stellenfleth, B. Djath, T. Neumann, S. Emeis, 2018: First in situ evidence of wakes in the far field behind offshore wind farms. Scientific Reports, 8, 2163. DOI: 10.1038/s41598-018-20389-y*

Answer: Thank you for pointing out this study. The results are extremely interesting. We added the reference to P2 L28 to L29 (P2 L26).

*(5) page 2, line 2: a reference is missing*

Answer: A link to copernicus.eu has been added on P3 L4 (P3,L2)

*(6) page 2, line 8: wind farm parameterisation in WRF: here, the parameterisation available in WRF by Fitch should be mentioned: Fitch, A. C., Olson, J. B., Lundquist, J. K., Dudhia, J., Gupta, A. K., Michalakes, J., & Barstad, I. (2012). Local and mesoscale impacts of wind farms as parameterized in a mesoscale NWP model. Monthly Weather Review, 140(9), 3017-3038.*

Answer: The reference has been added on P3 L11 (P3,L8)

*(7) How has the correlation coefficient displayed in Figure 2 been computed? From the bin means (black diamonds in the Figure) or from the full data set (grey circles in the Figure)? It actually does not look like that the grey circles can be explained by such a high correlation coefficient.*

Answer: The correlation has been calculated from the blue (gray if printed black and white) circles using the statistical tools implemented in the Software Mathematica 10.3. We did our own implementation of the correlation coefficient that confirms your suspicion of too high correlation coefficients. The problem seems to be a bug in the Mathematica software when forcing the fitting through the origin.

Figure 4 (2) and 5 (3) have been updated with the correct implementation of the correlation coefficient are now 0.74 and 0.78.

*(8) What is displayed in the last line of Table 3?*

Answer: The same as in line 2 but normalized with the wind speed at the center turbine of row A (turbine A15).

Changes: Caption of Table 3 P13 L7 to L8 (P11 L10)
$U_{15}$ has been changed to $U_{A15}$ to reflect more precisely that the wind speed at turbine location A15 is meant.

*(9) Subchapter 3.3.2: it does not really become clear why there are no discernible wake effects in the "short fetch" case.*

Answer: In order to determine wakes from SAR images before and after construction we need to assume that the wind conditions are similar between the two periods. This assumption is added more clearly to the manuscript. It is hard to draw a clear conclusions why the wake is not visible here. We made this uncertainty in the interpretation more clear.

Changes:
P13 L24 to L27 (P11,L19)
Removed (P15 L12)
P23 L14 to L18 (P20 L13)

*(10) Chapter 3.4 and Figure 11 (related to the preceding comment): Figs. 11 c and d are in contradiction to Figs. 8 b and 9. While there are clear wake effects in Figs. 11 c and d, there are none in Figs. 8 b and 9. The wind direction in both groups of figure is nearly identical. This contradiction should at least be addressed in the manuscript. This contradiction seems to be a clear hint that SAR data is not easily interpretable (see the above comment no. 2).*

Answer: We agree that there should be a better connection between these parts of the study. Additionally, we have added a more detailed explanation why transect b might be stronger affected by the differences in the

wind direction compared to transects c and d in P20 L7 to P21 L2. We agree that interpretation of SAR based wake is challenging, especially for fetch limited cases. We explicitly added the contradiction between Fig. 11 (9) and 13 (11) to motivate further studies.

Changes:
P20 L7 to P21 L10 (P17,L15 to P18,L2)
P23 L23 to L26 (P20,L18 to L21)

**Answer to anonymous Referee #2**

Dear Referee #2,

We would like to thank you for the suggested revisions to our manuscript and also for taking the time to point out technical corrections. The issues you raised were constructive and helpful to improve the quality of this paper and we have carefully considered them and revised this manuscript. Find enclosed our detailed answers to the comments that you raised.

Please note that the numbering of the Figures have changed since we added two new Figures. In our answers we refer to the updated Figure numbers and give the Figure numbers for the earlier version in brackets. Textual changes where we have considered your comments in the manuscript are given by page and line number for the new manuscript (not the one with track changes) and in brackets for the initial manuscript. Additionally, we have made a few changes where we felt the text could be improved and clarified.

Best regards,

Tobias Ahsbahs et. al.

**Comments regarding the methodology:**

*P4, from L16: Why select this GMF, these reanalysis data sets for derivation of SAR wind speed? Is this essentially describing how the DTU archive of wind speed maps is derived? If so, it would be clearer to turn this around to say that the archive of processed wind maps is used, and this archive is derived using SAROPS, CMOD5.N etc.*

Answer: This is correct remark. The aim in this paper was to use readily available SAR wind maps that can be downloaded from the Archive. We followed your suggestion and modified the text accordingly.

Changes: P5 L11 to L20 (P4 L15 to L24)

*P5, L23 : It is not clear what "averaging (WRF) wind direction at the same locations as for SCADA derived wind direction" means. Is this averaging across some time window or spatially, and to what end?*

Answer: Wind directions are spatially averaged using the same positions as for the SCADA wind direction for each time step. The sentence has been modified to clarify this.

Changes: P6 L21 to L22 (P5, L23)

*P6,Table 1: I am quite unclear on what 'Wind Direction' refers to. Isn't SAR wind speed using reanalysis data input all the time? Is this the wind direction used to select scenes, eg to determine that the wind direction is long or short fetch?*

Answer: Yes, this is the wind direction to select scenes. SAR wind retrievals are calculated using the wind direction specified under Section 2.2. We added an explanation to clarify this.

Changes: P7 L15 to L16 (P5,L26)

*P6, L9 : Why the hexagonal shape? How does it relate to SAR wind speed resolution cells?*

Answer: The hexagonal shape is chosen to include more resolution cells upstream in cross wind direction. This is inspired by footprint methods used in "Hasager, C.B. et al., 2004. Validation of ERS-2 SAR offshore wind-speed maps in the North Sea. International Journal of Remote Sensing, 25(191), pp.3817–3841.". Resolution cells are chosen if the center lies within the hexagonal shape. Additionally, when rotating the area to extract wind speeds this form allows including a wider angle of wind direction before resolution cells contaminated with reflections from the wind turbines are included.

Changes: P8 L7 to L14 (P6,L9 to 10)

*P10, L4 : Why is variation expected to be large? (This is kind of explained P11*
*L 14-15, but why not hypothesised here? It would be clearer.)*

Answer: This effect has been shown in previous studies (Peña, A. et al., 2017. On wake modeling, wind-farm gradients and AEP predictions at the Anholt wind farm. Wind Energy Science Discussions, 2017, pp.1–18. and Van Der Laan, M.P. et al., 2017. Challenges in simulating coastal effects on an offshore wind farm. Journal of Physics: Conference Series, 854(1).) and is caused by the roughness change between land and sea and the shape of the peninsula. We explicitly state this in the paper now.

Changes:
P11 L 13 to L15
P12 L 1 (P10,L4)

**Comments regarding treatment of Errors and Uncertainties:**

*P2,L10 : It is mentioned that the absolute accuracy of SAR derived wind speeds are low*
*compared to mast measurements. This needs further elaboration and consideration in*
*order to draw conclusions from the results.*

Answer: We added a paragraph to Sect. 2.2. giving a more detailed background on the accuracy of SAR wind retrievals, possible sources of uncertainties, and the ability to capture the mean wind speed. Usually the relative accuracy of SAR is much better than the absolute accuracy.

Changes:
(P2,L10) Sentence removed.
P3 L1 to L9 (P4L14)

*P8, L9 and P19, L3: Wind direction input is picked out as a source of error in deriving*
*SAR wind speeds, but is not explored. There is a passing reference to having also*
*looked at SAR winds derived with SCADA wind direction as input. This would seem to*
*offer an opportunity to explore and possibly quantify to some degree the uncertainties*
*arising from wind direction input.*

Answer: We have used the representative wind farm direction to retrieve wind speeds in Section 3.1.1. P10 L7 to 6 (P8 L7 to L9) states: "Using the wind direction from the SCADA system for the SAR wind retrieval process reduces the RMSE by approximately 0.1 m/s (not shown). This is a small improvement compared to the overall accuracy of the SAR wind retrieval process, thus supporting the SAR processing choice." We acknowledge that a more detailed study on the wind direction could be done, but feel that this is out of scope for this paper. We would prefer to keep the focus more on the wind speed variability and the wind farm wakes. We plan to conduct a more detailed study on the influence of wind direction on SAR wind retrievals using buoy data in the future.

*P5,L30: For SAR based wake studies it is assumed that turbines are operational. With*

*SCADA data available in parallel with SAR, the uncertainties arising from this assumption could be quantified allowing you to draw firmer conclusions from the results found.*

Answer: Overlapping data between the SAR wind fields and the SCADA data available to us are very limited. Only 47 scenes are at least partially overlapping with this period. It is therefore not possible to quantify how many of the wind turbines are operational. It would be possible to quantify this for the entire data set, but publishing this information is not possible under the None Disclosure Agreement with Ørsted. The overall turbine availability can be characterized as "high".

Changes: P7 L 18 to 20 (P5,L30)

*P5, L17; P7,L12; and P19, L9 and 17 : The simplification of adjusting between heights using a logarithmic wind profile is highlighted as a likely source of error. Please expand on this. What might be a better option, if you had information about the stability conditions? Uncertainties arising from this could be tested by trying other assumed atmospheric conditions and height adjustments and seeing to what degree they affect the delta U results. The discussion on p19 seems to contradict itself, suggesting that the height correction method should not be affected between upstream and downstream measurements as stability conditions will be similar for both (L9) but that downstream the logarithmic profile is no longer expected to be valid (L17).*

Answer: We have expanded on the influence of stability on the extrapolation and added this in a separate subsection, Sect. 2.5, and added a plot showing stability classes derived from WRF. We implemented stability correction for the extrapolation to test the influence of an altered stratification (assuming near stable and near unstable conditions). This has been done for results in Fig. 6 (4) where the resulting absolute wind speeds show difference of -0.4 m/s and 0.6 m/s, while the results relative to turbine position 15 show difference below 0.01%.
Regarding the contradiction pointed out: The argument for similar stratification was intended for the wind speed variability, Fig. 6 (4), upstream in a region unaffected by the wake.

Changes:
Add Sect. 2.5 P6 L24 to P7 L10
P9 L13 to L17 (P7 L8 to L13)
P10 L8 to L9 (P8,L10)
P13 L16 to L22 (P11 L 16)
P22 L12 to L15 (omitted P19,L6 to L12)
P22 L26 to L28

*P10,L25 : An anomalous result at position A05 is dismissed as a problem and not real, but can this be justified. Can the SCADA data reveal evidence for this assertion?*

Answer: The applied method for calculating the equivalent wind speed fails when the operational behaviour of the wind turbine differs from the expectation. Then, the derived wind speed is not reliable any longer. A more close look into the data revealed that the SCADA derived wind speed at turbine A05 is higher for wind speeds above 12 m/s and lower for wind speeds bellowed compared to the neighbouring turbines. We are still uncertain of the exact reason of this deviation but did not want to publically speculate. We have addressed this problem in Section 2.3 and dismissed data from turbine A05 from the subsequent analysis in Sect. 3.1 and 3.2.
Changes: P6 L3 (P10,L25)

**Comments regarding the conclusions drawn from the results:**

*P9,L11 : I am not satisfied by the suggestion that the lower RMSE seen downstream*

*might be due to sample size. Lower RMSE would seem to imply that the SAR 'model '*
*data better fits the SCADA observations downstream.*

Answer: We acknowledge that the sentence might be misleading as to the cause of the difference in the RMSE. The respective sentence has been modified to make it more neutral and the issue is picked up in the Discussion Section where we suggest further studies are needed to investigate this behaviour.

Changes: P22 L26 to L30 (P18,L29)

*P21, L14: The wakes study with SAR does appear to show a wind farm wake effect in*
*the long fetch scenario, but I am not satisfied that this allows the conclusion that 'strong*
*indications of wind farm wake effects: : :' . This is because the short fetch scenario*
*shows no, or weak effects, and in the long fetch scenario there appears to be some*
*unexplained wind speed up after the wind farm is constructed in the region north of the*
*island (P14,figure 7). I think it can only be concluded that it's a promising technique*
*and should be revisited in the future with more post wind farm SAR data.*

Answer: This is a valid point. We suggest to modify the text to be more conservative in the conclusions drawn from the results and add a sentence on the necessity of more studies in the future. Additionally, we expanded the discussion on possible reasons why wind farm wakes are harder to detect for short fetch cases.

Changes:
P23 L11 to L 26 (P20 L11 to L21)
P24 L21 to L23 (P21,L14 to L17)

*P17,L15 : The explanation for the deviations between before and after upstream wind*
*speeds in transect b is not at all satisfactory. There is no explanation of why transect*
*b might be more sensitive to wind direction than say d. This theory should be further*
*explored by considering any bias in the wind direction distribution between the before*
*and after wind farm images and comparing this to the transect locations relative to the*
*coast.*

Answer: We agree that the argument made here was too short and not comprehensive enough. We expanded the argument on the influence of changes in the fetch between the different transects more specifically. Since the wind direction is taken from models, the local wind direction on the transect will in general be different. Transect b is located close to the Northern side of the peninsula. A higher occurrence of more Westerly wind direction after the construction of the wind farm would affect the fetch on transect b more compared to transect c and d. The distribution of wind direction is unfortunately not available from in situ measurements for the considered periods, so we cannot certainly determine the deviations in the modelled wind direction.

Changes:
P20 L7 to P21 L10 (P17,L15 to P18,L2)
P23 L23 to L26 (P20,L18 to L21)

**Technical Corrections:**
*P1, L20 : Give the date that these wind capacity numbers are valid for.*

Answer: The numbers where for 2016, but a new report for 2017 is now available. The number have been updated accordingly and the reference year been added.

Changes: P1,L20 (P1,L20)

*P2, L14,18 : Insert brackets around these references (name and year) for consistency with others in the paper.*

Answer: For the references mentioned the reference itself is the subject or object of the sentence and was therefore not put in brackets. This is consistent with the WES style guide (https://www.wind-energy-science.net/Copernicus_Publications_Reference_Types.pdf) and other papers published in this journal.

*P2 , L6 : "wind is causing" should be "wind causes".*
Answer: Change made

*P3, L2 : Missing reference*
Answer: Link added

*P4, L6 : Suggest inserting ", and was " between 400MW and constructed to make the sentence clearer.*
Answer: Change implemented

*P4, L16: Insert "," after wind speed retrieval.*
Answer: Change implemented

*P5, L5: Explain "curtailed" for the benefit of readers from the remote sensing rather than energy community.*

Answer: Explanation has been added:
Changes: P6 L1 to L2 (P5, L5)

P5,L8 : "on at edge" should be "on the edge" or "at the edge".
Answer: Answer: Change implemented to "on the edge"

*P11,L6 : Number the equation and then reference it in Table 3.*
Answer: Equation number added and referenced in caption of Table 3

*P13, L18 : "with on standard error" should be "with one standard error".*
Answer: Change implemented

*P16, Figure 10 : The turbines marked in black are not clear to me.*
Answer: The turbines lying within the transects are shown in black. A legend has been added to the respective Figure.

*P18, L11 : "a as" should be "as a".*
Answer: Change implemented.

*P19, L24: "is experiencing" should be "experiences".*
Answer: Change implemented.

*P20: L7: "approx." should be "approximately".*
Answer: Change implemented.

**Applications of satellite winds for the offshore wind farm site Anholt**

Tobias Ahsbahs[1], Merete Badger[1], Patrick Volker[1], Kurt S. Hansen[1], Charlotte B. Hasager[1]

[1]Department of Wind Energy, Technical University of Denmark, Roskilde, 4000, Denmark

*Correspondence to*: Tobias Ahsbahs (ttah@dtu.com)

**Abstract.** Rapid growth in the offshore wind energy sector means more offshore wind farms are placed closer to each other and in the lee of large land masses. Synthetic Aperture Radar (SAR) offers maps of the wind speed offshore with high resolution over large areas. These can be used to detect horizontal wind speed gradients close to shore and wind farm wake effects. SAR observations have become much more available with the free and open access to data from European satellite missions through Copernicus. Examples of applications and tools for using large archives of SAR wind maps to aid offshore site assessment are few. The Anholt wind farm operated by the utility company Ørsted is located in coastal waters and experiences strong spatial variations in the mean wind speed. Wind speeds derived from the Supervisory Control And Data Acquisition (SCADA) system are available at the turbine locations for comparison with winds retrieved from SAR. The correlation is good, both for free stream and waked conditions. Spatial wind speed variations along the rows of wind turbines derived from SAR wind maps prior to the wind farm construction  agree well with information gathered by the SCADA system and a numerical weather prediction model. Wind farm wakes are detected by comparisons between images before and after the wind farm construction. SAR wind maps clearly show wakes for long and constant fetches but the wake effect is less pronounced for short and varying fetches. Our results suggest that SAR wind maps can support offshore wind energy site assessment by introducing observations in the early phases of wind farm projects.

**1    Introduction**

Europe now has a total installed offshore wind capacity of 15,780 MW (status as end of 2017) corresponding to 4,149 grid-connected wind turbines across 11 countries. By 2020, offshore wind is projected to grow to a total installed capacity of 25 GW (Wind Europe 2018).

In Northern Europe much of this development is happening in the North Sea and the Baltic Sea. With an increasing amount of wind farms already erected, suitable locations with prevailing wind directions undisturbed by land or other wind farms are becoming scarce. Therefore, new wind farms are built in less favourable locations e.g. in the lee of land masses or large wind farms. Additionally, many shore lines are not straight but have a complex geometry that is determined by peninsulas, bays and islands. The lee effect of land i.e. the horizontal wind speed gradient due to a varying distance to shore (fetch) and wind

farm wakes from other wind farms both influences the wind resource. Dörenkämper et al. (2015) found that large "horizontal streaks of reduced wind speeds that under stable stratification are advected several tens of kilometres over the sea" can severely affect offshore wind farms. Correct prediction of the wind resource influenced by either land or adjacent wind farms, or a combination of the two, is a challenging problem. This study is motivated by this challenge and focuses on the Anholt offshore wind farm in the Kattegat Strait in Denmark. It involves analysis of satellite-based Synthetic Aperture Radar (SAR) wind maps, wind turbine data, and simulation results from the Weather Research and Forecasting (WRF) model.

[revised manuscript text omitted]

5   SAR wind retrievals are indirect estimates of the wind speed that rely on the assumption that a measurement of the radar backscatter from the sea surface can be converted to a corresponding wind speed at the height 10 m. This is possible because the SAR observations are sensitive to cm-scale waves at the sea surface, which are generated by the instantaneous wind stress. Phenomena that modify the small-scale ocean surface waves i.e. biological or mineral films (Gade & Alpers 1997), and sea states (Alpers et al. 1981) influence the wind speed retrieval. This adds uncertainties to the wind speed retrieval.

10  Global validation studies of satellite wind retrievals against modelled wind speeds found RMSE of 1.30 m/s (Hersbach 2010) while validations against in situ measurements in the Baltic showed an RMSE of 1.17 m/s (Hasager et al. 2011). Both studies show that while the accuracy of individual wind speed retrievals is somewhat low, SAR wind fields capture the mean wind speed and its spatial variability well.

15  An archive of processed wind maps from Envisat and Sentinel-1 A and B over Europe is available from DTU Wind Energy[2]. Our analyses are based on these readily available SAR wind maps.  In the archive, wind speeds are retrieved from the SAR scenes using the SAR Ocean Products System (SAROPS) (Monaldo et al. 2015). The GMF called CMOD5.N (Hersbach 2010) is chosen for the wind speed retrieval, and wind directions are needed as an ancillary input for processing. We obtain the wind directions from the Climate Forecast System Reanalysis data set (CFSR[3]) during 2002-10 and from the

20  Global Forecasting System (GFS[4]) during 2011-17. To reduce effects of random noise in the SAR imagery and to smooth out effects of longer period waves that modify the local radar incidence angle,  the SAR scenes are resampled to 500-m pixel size in connection with the wind retrieval processing. Hard targets like wind turbines or offshore substations cause a strong signal in SAR images. The increased backscatter signal will cause an overestimation of the retrieved wind speed and therefore, extremely bright resolution cells are filtered out of the SAR wind maps prior to our analyses.

25
* * *
[2] https://satwinds.windenergy.dtu.dk https://satwinds.windenergy.dtu.dk
[3]  http://nomads.ncdc.noaa.gov/data.php?name=access#cfs-reanal-data
[4]  http://nomads.ncdc.noaa.gov/data/gfsanl
[5]  https://satwinds.windenergy.dtu.dk

**2.3    SCADA data**

SCADA systems monitor record wind turbine data, i.e. power production or pitch angle. The wind turbine power curve links the free wind speed to a power production. This wind speed (hereafter SCADA wind speed) can be derived from power and pitch combined with the power curve provided by the turbine manufacturer. The power is monotonically increasing with the wind speed between cut-in and rated power. Therefore, the wind speed can  be inferred for this region. From rated power to cut-out, the power is constant but the blades are pitched increasingly. For this region, the wind speed can be inferred from the pitch signal. The resulting wind speed is equivalent to the reference wind speed used to create the power curve and is treated as a measurement at hub height. An inter-comparison between the turbines reveals that this is - in general an acceptable approach.

A qualification procedure is used to eliminate periods where the wind turbines are not grid connected and are not producing power during a complete 10-minute period or have been curtailed, meaning their power generation has been reduced. Unfortunately, the wind speed for turbine A05 deviates due to unknown reasons and will be excluded from the analysis. The remaining periods are applicable for analysis after a final examination of the power curve. Due to a lack of undisturbed mast measurements, the inflow conditions need to be derived from the operational wind turbine data themselves. The inflow reference wind direction is determined from calibrated, undisturbed selected wind turbine yaw positions on the edge of the wind farm (cf. Peña et al. (2017) for further details).

**2.4    Numerical wind simulations**

The numerical simulations used in this study are performed with WRF version 3.5 without wind farm parametrization. The total simulated period covers 28 years from 1990 to 2017. Simulations are performed in 10-day chunks. Each individual simulation extends in total over 11 days, with the first day being disregarded as a spin-up period. The computational domain consists of three nests with an 18 km, 6 km, and 2 km grid spacing, respectively. Here the outermost domain is forced by (ECMWF) ERA-Interim Reanalysis (Dee et al. 2011) and the results of the inner-most domain are used for the analysis. In the horizontal direction, the innermost domain extends over 854 km and 604 km in the x and y direction. In the vertical direction, 41 vertical levels with model top at 50hPa are used, with 9 levels being within 1000 m from the surface. Wind speeds at the turbine hub height are derived by logarithmic interpolation between the two closest model levels.

The most relevant physics parametrizations in the model set-up, are the Yonsei University Scheme (YSU) Planetary Boundary Layer (PBL) scheme (Hong et al. 2006) and the MM5 similarity surface-layer scheme, and Sea surface temperatures from NOAA/NCEP  (Reynolds et al. 2010). Further details of the model set-up and its validation are given in Peña & Hahmann (2017). WRF wind directions at the same locations as for the SCADA derived wind direction are extracted and averaged to a time series of representative wind directions.

**2.5    Wind speed extrapolation**

SAR wind speeds are retrieved for a height of 10 m and SCADA wind speeds are representative for the wind turbine hub height at 81.6 m. A wind profile needs to be applied to perform wind speed extrapolation between these two levels. Ideally, the local stratification should be considered but no measurements that could lead to a quantification of atmospheric stability effects are available on site. In lack of local measurements, we assume a logarithmic wind profile with a wind speed dependent roughness length using Charnock's relation and the Charnock parameter (Grachev & Fairall 1996). Later we test this assumption through a sensitivity analysis using stability correction to the logarithmic wind profile (Wyngaard 2010, p 222).

The (Peña & Hahmann 2017). WRF wind directions representative for the wind farm domain are calculated by averaging wind direction at the same locations as for the SCADA derived wind direction. WRF model outputs include stability information expressed as the length scale $z/L$. We use this to investigate the frequency of occurrence for different stability classes. Stability information from WRF is not sufficiently accurate to perform a stability correction of the wind profile for individual SAR samples (Badger et al. 2016). Stability classes at the turbine hub height are defined using the definitions from Hansen et al. (2012) for WRF simulations coinciding with the SCADA time series at the wind farm location, see Figure 2. The WRF outputs indicate that neutral stratification is increasing with the wind speed and the overall distribution favours stable stratification over unstable at the Anholt site. These findings differ from simulation and measurements of stratification in the Baltic, which favour stable stratification (Smedman et al. 1997).

[Figure]

**Figure 2: Distribution of 7 stability classes from very stable (vs) to very unstable (vu) based on 2.5 years of WRF simulations for wind speeds between 4 m/s and 20m/s (left) and the distribution for all wind speeds (right).**

**3 Methods & Results**

Four different methods are applied to analyse SAR wind fields around the Anholt wind farm. These are listed in Table 1 together with the temporal coverage for SCADA, SAR, and WRF data used in the respective analysis. The SCADA winds are used as reference measurements. "Wind direction" specifies the data input used for selection of SAR wind fields in contrast to the wind direction used for the SAR wind retrieval described in Sect 2.2. Averaged wind speeds can show strong gradients in two directions.  In the following, the term 'wind speed gradient' refers to wind speed changes perpendicular to the coastline whereas the term 'wind speed variability' refers changes along the  rows of wind turbines. For SAR based wake studies in Sect 3.3 and 3.4 we assume that all turbines at Anholt are operational. Data about the overall turbine availability is not available for publication for proprietary reasons.

Table 1: Overview of the data sets and time periods used for the analysis.

[revised manuscript text omitted]

$$\Delta U_{N,S} = \sum_{i=A28}^{A31} U_i - \sum_{i=A01}^{A03} U_i$$

$$\Delta U_{N,S} = \sum_{i=A28}^{A31} U_i - \sum_{i=A01}^{A03} U_i \quad\quad\quad\quad\quad\quad\quad\quad\quad\quad\quad (1)$$

Where $U_i$ is the mean wind speed at the turbine location. The difference between the Northern and the Southern part of the wind farm is given in Table 3. SCADA and SAR agree within 0.1 percentage point while WRF predicts a 1 percentage point larger difference than SCADA results suggest.

Table 3: Sample sizes, difference between wind speed at the most Northern and Southern turbines $\Delta U_{N,S}$ (three turbine location averaged, see Equation 1), and the same difference normalized with the wind speed at turbine $U_{A15}$ at turbine A15.

|  | SAR | WRF SAR | WRF | SCADA |
|---|---|---|---|---|
| Samples N [-] | 72 | 72 | 10524 | 4625 |
| $\Delta U_{N,S}$ [m/s] | 0.92 | 1.02 | 0.98 | 0.95 |
| $\Delta U_{N,S}/$ $U_{A15}$[%] | 8.8 | 10.3 | 9.8 | 8.7 |

The wind speed variability along Row A, as shown in Figure 6 and Table 3, is likely caused by varying fetch from the coastline of Djursland. The fetch at different positions along Row A can vary between 16 km and 50 km for the same wind direction, see Figure 1. The agreement between nondimensional wind speeds from SAR and SCADA is remarkably good. We can conclude that for this site, wind speeds retrieved from SAR imagery could have predicted the relative wind speed gradients well, before construction of the wind farm.

We test the influence of extrapolation by assuming the turbine hub height is within the surface layer and that both the atmospheric stability and the aerodynamic roughness length are constant along turbine row A. The relative wind speed should thus show little dependence of the height since the stability correction term has the same value. This has been tested assuming near stable and near unstable conditions. The resulting extrapolated wind speed (not shown) differs between -0.4 m/s (unstable) and 0.6 m/s (stable), while the results relative to turbine position 15 show differences below 0.01 percentage points. These assumptions will not be valid at all times, but the extrapolation error of the mean wind speed from 10m to hub height is expected to be reduced when the mean wind speed is divided by the mean wind speed at a reference location.

**3.3    Wind farm wakes from SAR**

To investigate the impact of the Anholt wind farm on the wind conditions in the area, we compare wind speeds extracted from SAR wind maps along two transects before and after wind farm construction. With this approach, a baseline of wind conditions before wind farm construction can be determined assuming that the wind conditions in the period before and after the wind farm construction are similar.

Wind farm wakes at Anholt are analysed for two wind direction sectors. The first sector (75°-105°) represents easterly wind directions and a long fetch. The second sector (255°-285°) represents Westerly wind directions and a short fetch, see Figure 1. Wind direction information from WRF is used for the selection of SAR wind maps within the two sectors as described in Sect. 2.4.  
[revised manuscript text omitted]
,  whereas larger deviations are found at transect b.  These deviations might be caused by variations of the fetch. Wind speed extracts along transect b are likely to be very sensitive to the local wind direction because transect b is located close to the Northern side of the peninsula Djursland. Here, a

10 small change in the wind direction could lead to a large increase or decrease of the fetch, see Figure 12. An increase of the fetch is usually associated with an increase in the wind speed. Therefore, a higher occurrence of wind directions West of 235° after  construction of the wind farm could be the reason for the deviations observed for transect b. Transect c and d would be less affected by variations of the wind direction since they are located further South where the fetch varies less for Southwesterly wind directions. The wind

15 direction used for the selection of SAR images comes from WRF simulations at the wind farm location. Any local variability

of the wind direction is not resolved by WRF and the true wind direction along the four transects might thus deviate from the WRF wind direction. Since we do not have in situ measurements for the entire period considered here, it is not possible to determine the exact difference in the wind direction distribution.

Wind speeds downstream of the wind farm show a positive wind speed gradient along  transects b, c, and d. Here, the wind speed on transect b is similar before and after wind farm construction. This transect crosses a narrow part of the wind farm with only three turbine rows. Transects c and d cross a larger number of turbines and show a significant change of the wind speed after the wind farm construction and we attribute this change to wake effects of the wind turbines.

SAR wind speeds cannot be retrieved correctly within the wind farm itself due to radar reflection from the turbines. The SCADA wind speeds for turbines within transect b to d are used instead to describe the wind speed behaviour within the wind farm. The SCADA wind speeds suggest a reduction of wind speeds downstream of turbine row A which is most pronounced for transect c and d crossing many turbine rows.

SCADA wind speeds show the wind farm wake  as a reduction in wind speed compared to the upstream turbine. SAR winds on transect c and d show a reduction of wind speed compared to the situation before construction of the wind farm. The deviations between these two types of wind speed information are between 0.3-0.6 m/s. Differences between SAR and SCADA winds may be attributed  to i) a difference in the location with SCADA winds at the turbine positions and SAR winds downstream of the wind farm, ii) differences in the sample size and measurements that are not collocated in time, or iii) differences in the vertical position of the measurements. SCADA data are derived at the turbine operating height whereas the SAR wind retrievals are based on observations of the sea surface. The strongest wind turbine wake effect is expected at the turbine hub height, which is consistent with a stronger wake from SCADA winds compared to SAR.

**4    Discussion**

We have demonstrated how an extensive archive of SAR wind maps can be used to identify the combined effects of a complex coastal geometry and wind farm wakes on the mean wind conditions around the Anholt wind farm. Our results illustrate how wind maps retrieved from SAR can predict the wind conditions that offshore wind turbines and whole wind farms experience before a wind farm is constructed.

For the first time, wind speeds derived from the SCADA system of an entire wind farm have been compared to SAR wind speeds, see Figure 4. The correlation for free stream conditions is good and the slope of the fit is very close to one. This result is encouraging for using SAR derived mean wind speeds to predict wind conditions as experienced by the wind turbines. GMFs used for SAR wind retrieval are tuned using observational data from buoys in the open ocean. Influences of

internal boundary layers caused by the roughness change between land and sea, or effects of limited fetch on the ocean surface roughness are not fully accounted for. These effects are hard to quantify, but the RMSE compared to lidar measurements in the coastal zone is between 1.3 and 1.4 m/s (Ahsbahs et al. 2017). The SAR wind speed retrieval process needs a wind direction as an input. Readily available SAR wind maps using a global model wind direction are used throughout this study. –Therefore, uncertainties in the modelled wind direction translate into errors in the wind speed retrievals.

The Anholt wind farm experiences strong variability in the wind speed along the Western-most row (Row A) for the prevailing wind directions from 245-275°. Comparisons of WRF mean wind variability from the full time series with a downsampled data set matching 72 SAR images before construction show similar results. This strengthens the assumption that the available SAR images correctly represent the mean wind conditions at the Anholt site. The normalized mean wind speed obtained from SAR before construction of the wind farm agrees very well with results from SCADA winds of the first 2.5 years of wind farm operation. The mean wind speed between the South and North of row A increases by 8.7% in the SCADA wind speeds and 8.8% in SAR derived wind speeds, see Table 3. SAR wind maps are thus valuable for characterization of large scale flow phenomena such as wind speed variations over long rows of turbines. Variability in the wind speed relative to a reference location is expected to show little influence from atmospheric stability as presented in Sect. 3.2. The validity of this claim hinges on assumptions of surface layer theory, constant roughness, and stability over the domain. A more detailed study to test these assumptions could support the use of SAR for detection wind speed variabilities.

For this site, nondimensional wind speeds from WRF at the turbine locations also predict wind speed variability very similar to results from SAR and SCADA. Models such as WRF are powerful tools to identify good wind resources, but cannot fully replace observations of the wind conditions on site. The presented analysis of SAR wind maps can complement modelling efforts by introducing an independent measurement for comparison, since both data sets are available before construction of a potential wind farm. The wind retrieval process assumes a logarithmic wind profile. Influences of atmospheric stability on the instantaneous comparison between SAR and SCADA wind speeds cannot be accounted for without site specific measurements. To overcome this problem, SAR wind speeds can be presented relative to a reference location as shown in our analyses. Assuming Monin-Obukov theory, constant stability, and roughness over the domain introduces a stability correction factor that is independent of the location and height. The relative wind speed is thus independent of height. These assumptions will not be valid at all times, but the extrapolation error of the mean wind speed from 10m to hub height is expected to decrease when the mean wind speed is divided by a reference location.

A good agreement between WRF and SAR with regard to wind speed variability can add confidence to wind resource assessment. Further studies at locations where the mean wind speed is affected by an upstream shoreline could show if agreement between SAR and NWP modelling is common and if disagreements could point towards an increased uncertainty in the NWP modelled wind resources.

Comparisons in the wake (see Figure 5) showed a lower scatter than free stream comparisons suggesting a better fit in waked compared to free stream conditions, even though the assumptions of a fully developed wind profile are violated by the presence of a wake. Further studies of SAR wind retrievals within wind farm wakes using high quality reference measurements at several heights from the sea surface to the turbine hub height are needed in to examine this finding in more detail.

The correlation of

 atmospheric boundary layer changes significantly with the presence of wind farms. This will affect our comparisons of SAR and SCADA wind speeds downstream comparison is good but the bias towards higher wind speeds from SAR has increased compared to the analysis upstream of the wind farm, see Figure 4 and Figure 5. The largest wake deficit is located at hub height (Porté-Agel et al. 2011) and this could cause an overprediction of the SAR wind speed when extrapolated in Figure 5. Additionally, SAR winds are retrieved between 600m and 2600m downstream of the turbine position but are compared to SCADA wind speeds at the turbine location and the wake is likely to recover. This is also consistent with the difference between SAR and SCADA winds in Figure 13. To better quantify wind farm wakes from SAR images, further work is needed to understand how wakes interact with the ocean surface and how this influences SAR wind retrievals.

~~The Anholt wind farm is experiencing strong variability in the wind speed along the westernmost row (Row A) for the prevailing wind directions from 245-275°. The normalized mean wind speed obtained from 72 SAR images before construction of the wind farm agrees very well with results from SCADA winds of the first 2.5 years of wind farm operation. The mean wind speed between South and North of row A increases by 8.7% in the SCADA wind speeds and 8.8% in SAR derived wind speeds, see Table 3. SAR wind maps are valuable for characterization of large scale flow phenomena such as wind speed variations over long rows of turbines.~~

~~Nondimensional wind speeds from WRF at the turbine locations also predict wind speed variability very similar to results from SAR and SCADA. Models such as WRF are powerful tools to identify good wind resources, but cannot fully replace observations of the wind conditions on site. The presented analysis of SAR wind maps can complement modelling efforts by introducing an independent measurement for comparison, since both data sets are available before construction of a potential wind farm.The good agreement between WRF and SAR with regard to wind speed variability adds confidence to assessments of the wind resource.~~

Anholt wind farm has irregular turbine spacing and the shape is elongated. Methods applied at other offshore wind farm sites for analysing wakes in SAR wind maps (Hasager, Vincent, et al. 2015) are less suitable for Anholt . A new approach for analysing wind farm wakes from SAR images has therefore been suggested here, which explores the difference of SAR wind maps before and after the wind farm was constructed. The wind farm wake effects are analysed

5 along transects approximately perpendicular to the wind direction on the upstream versus the downstream side of the wind farm and along transects crossing the wind farm aligned with the wind direction.

For situations with a long fetch, perpendicular transects before wind farm construction provide a suitable baseline to check averaged differences between upstream and downstream transects, see Figure 9. The wind farm wake estimated

10 from SAR shows a structure that roughly follows the turbine density of the wind farm. In contrast, no indication of a wake is found in Figure 11. The wind direction sector overlaps with the sector from Figure 6 where strong horizontal wind speed variability was found, which will also affect the transects. A possible explanation could be that the upstream orography is more complex for the short compared to the long fetch scenario. This could affect the similarity of the wind conditions for the SAR images

15 before and after wind farm construction, either due to difference in the wind direction or atmospheric stratification.

Transects crossing the wind farm can be used  to investigate how the coastal wind speed gradient and wakes of the wind farm interact, see Figure 13. No wind speed reductions compared to the upstream reference point are found.  but

20 two transects going through an area of high wind turbine density show a reduction of the mean wind speed after wind farm construction compared to the situation before. This results stands in contrast to Figure 11 and transect b in Figure 13.  where no evidence of a wind farm wake was found. Identification of wakes

25 from SAR images is not trivial when an upstream coastline is influencing the flow. Further studies at locations with simple geometry of the coastline would help to understand the interplay of wind farm wakes and coastal wind speed gradients.

SAR wind maps are suitable for analysing large scale wind conditions and they can show the combined effects of different flow phenomena. In this analysis, wind farm wakes, coastal wind speed gradients, and wind speed variability from differing

30 fetch occur simultaneously. It is challenging to identify the contribution of one particular flow phenomenon, e.g. wind farm wakes from this data. In contrast to engineering wake models such as FUGA or Park that are run with a single wind speed and direction, SAR wind maps capture the full picture of the flow around a wind farm. The presented methods can potentially be repeated for any offshore wind farm site even before the wind farm construction.

The presented SAR data archive goes back to 2002 and offers the possibility of reference measurements before most of the current offshore wind farms were constructed. The analyses presented in this study will gain confidence as the satellite data archives are growing over time. With Sentinel-1 A and B, two new satellites are acquiring new scenes on a daily basis which are available in the public domain. This makes SAR observations and derived wind maps more accessible and the time is

5 right to develop tools for SAR data analysis that are tailored to the needs of the offshore wind industry.

**5    Conclusion**

Large archives of SAR wind maps have recently become publically available and the sampling frequency of the measurements has increased significantly with the European SAR missions Sentinel-1 A/B. Readily available SAR based wind speed maps represent a computationally and monetarily cheap source of information about the large scale wind speed

10 variability offshore. The maps are available in hindcast and may thus be used from the earliest stages of a wind farm project. We have demonstrated that wind speed maps retrieved from SAR observations of radar backscatter can be used to predict the spatial wind speed variability at a potential wind farm site before construction begins. The satellite based wind speed maps can also be used for characterization of wake effects around existing wind farms and to partially determine the cumulative effects of coastal wind speed gradients and wake effects.

Wind speeds retrieved from SAR correlate well with the SCADA derived wind speeds for the turbines at Anholt wind farm. RMSEs are 2.23 m/s and 2.12 m/s for comparisons upstream and downstream of the wind farm, respectively. Wind farm wakes are detected from SAR wind fields using a long time series with measurements before and after construction of the wind farm. This approach is promising, since a baseline of wind conditions before the construction is available.

20 Wind farm wake effects are found for wind directions leading to a long fetch with a maximum deficit of 0.7m/s. Wind farm wakes at fetch limited conditions are harder to identify possibly due to the complex interplay of different effects such as varying fetch and coastal wind speed gradients on the mean wind speed. More studies using these approaches for different wind farms are necessary, ideally with in situ reference measurements, to determine the capabilities of SAR for wind farm wake detection.

Our results  that SAR wind maps can resolve smaller-scale wind variability comparable to SCADA wind speeds . WRF and SAR data sets are independent of each other and are available in the  early stages of planning an offshore wind farm. Alongside with model simulations, satellite based wind maps

30 represent a valuable resource to introduce large scale on-site measurements early in an offshore wind farm project, i.e. for planning of on situ measurement campaigns.

**Data availability:**

SAR wind fields are available at https://satwinds.windenergy.dtu.dk/ and WRF model runs can be made available upon request. SCADA data is not available for publication.

**Author contribution:**

5  Tobias Ahsbahs developed methods and code. Merete Badger and Charlotte B. Hasager provided the processed SAR wind maps and contributed with guidance and comments. Kurt S. Hansen prepared the SCADA data and Patrick Volker provided the WRF data. Tobias Ahsbahs prepared the manuscript with contributions from all co-authors. This work is part of Tobias Ahsbahs' PhD under supervision of Merete Badger.

**Competing interests:**

10 The authors declare that they have no competing interest.

**Acknowledgements:**

We would like to acknowledge Ørsted for granting access to data from the Anholt wind farm, Johns Hopkins University Applied Physics Laboratory and the National Atmospheric and Oceanographic Administration (NOAA) for the use of the SAROPS system, and ESA for providing public access to data from Sentinel-1A. Personal thanks to Nicolai G. Nygaard
15 from Ørsted for his approval and comments.